# Towards Truly Zero-shot Compositional Visual Reasoning with LLMs as Programmers

**Aleksandar Stanić**[*]                                   *alexstanic@google.com*
*Google DeepMind*

**Sergi Caelles**                                          *scaelles@google.com*
*Google Research*

**Michael Tschannen**                                     *tschannen@google.com*
*Google DeepMind*

**Reviewed on OpenReview:** *https://openreview.net/forum?id=WYGiqSVstK*

## Abstract

Visual reasoning is dominated by end-to-end neural networks scaled to billions of model parameters and training examples. However, even the largest models struggle with compositional reasoning, generalization, fine-grained spatial and temporal reasoning, and counting. Visual reasoning with large language models (LLMs) as controllers can, in principle, address these limitations by decomposing the task and solving subtasks by orchestrating a set of (visual) tools. Recently, these models achieved great performance on tasks such as compositional visual question answering, visual grounding, and video temporal reasoning. Nevertheless, in their current form, these models heavily rely on human engineering of in-context examples in the prompt, which are often dataset- and task-specific and require significant labor by highly skilled programmers. In this work, we present a framework that mitigates these issues by introducing spatially and temporally abstract routines and by leveraging a small number of labeled examples to automatically generate in-context examples, thereby avoiding human-created in-context examples. On a number of visual reasoning tasks, we show that our framework leads to consistent gains in performance, makes LLMs as controllers setup more robust, and removes the need for human engineering of in-context examples.

## 1 Introduction

Compositional visual question answering requires a model to answer questions about visual content in a compositional manner, involving multiple steps of reasoning or considering relationships between different objects or entities within an image. It is a complex task as it requires understanding both the visual information in an image and the structure of the question, and reasoning about how different visual elements relate to one another to generate the correct answer. Recently, large progress has been made on many such vision and language tasks by scaling end-to-end neural networks models in terms of size, training data, and compute (Alayrac et al., 2022; Chen et al., 2022b; Yu et al., 2022; Wang et al., 2022a; Gan et al., 2022; Lu et al., 2022; Li et al., 2023b; Driess et al., 2023; Chen et al., 2023d;c). However, even the largest state-of-the-art (SotA) models struggle in tasks that require compositional reasoning, ability to generalize, fine-grained spatial reasoning capabilities, and counting (Bugliarello et al., 2023; Paiss et al., 2023; Hsieh et al., 2023; Yuksekgonul et al., 2022; Zhao et al., 2022; Hendricks & Nematzadeh, 2021). An example of such task is the following query: "Could the cookies on the table be equally distributed among children?" (Surís et al., 2023). To solve this, the model needs to detect the cookies in the image, filter out the ones that are not on the table, detect children, count cookies and children, and check if the cookies count is divisible

---

[*]Work completed during an internship at Google Research.

by the children count. Questions like these are difficult for current end-to-end vision and language models (VLMs). Scaling VLMs further makes them even more data- and compute-hungry(Villalobos et al., 2022), so the scale alone seems unlikely to solve these tasks, especially due to the exponentially-large long tail of compositional tasks.

On the other hand, it is questionable whether solving compositional tasks with a single monolithic end-to-end neural network is the optimal approach. Intuitively, it might be easier to first decompose the task into several subtasks, individually solve the subtasks, and then use the intermediate results to solve the original task. This is reminiscent of the way humans approach compositional problems. According to Daniel Kahneman's framework (Kahneman, 2017), our thought process can be thought of as consisting of two mechanisms: System 1 and System 2. System 2 is the "slow", "analytical" system that can decompose the task into subtasks, while System 1 is the "fast", "reactive" system that solves individual tasks such as recognizing patterns. In machine learning, the early work on task decomposition was pioneered by Neural Module Networks (NMNs) (Andreas et al., 2016; Johnson et al., 2017; Hu et al., 2017). NMNs are trained end-to-end in the hope that every module will learn a separate function that will be reusable across tasks. However, these models have a number of drawbacks, namely that the program generation requires hand-tuned parsers, they are difficult to optimize (sometimes requiring reinforcement learning), and they have the issue of a module "collapse", where some modules are never activated and others take over all the work, contrary to the design intentions.

Recently, an alternative approach based on "tool use" gained popularity (Cobbe et al., 2021; Komeili et al., 2021; Thoppilan et al., 2022; Parisi et al., 2022; Zeng et al., 2022; Gao et al., 2023a; Qin et al., 2023; Zhuge et al., 2023). In "tool use", an LLM solves a task by controlling (akin to System 2) a set of tools (such as an object detector, akin to System 1) (Zeng et al., 2022; Shen et al., 2023; Zhuge et al., 2023). In particular, VisProg (Gupta & Kembhavi, 2023), ViperGPT (Surís et al., 2023), and CodeVQA (Subramanian et al., 2023) show great promise in solving visual question answering by using an LLM to generate a program (in Python or a custom scripting language). During execution, the program calls individual vision models (such as object detector, depth estimator) through an *API* that is provided in the prompt. For example, to answer "Which color is the jacket of the second person from the left?" (Figure 1a), the program needs to detect people, sort them from left to right, select the second, detect their jacket, and query its color. These models achieved SotA on compositional visual question answering, visual grounding, and video temporal reasoning tasks. By their construction, they are interpretable, compositional, adaptable (tools can be upgraded on the fly), offer strong generalization, mathematical, and reasoning skills, and do not require gradient-based training. However, in their current form, they heavily rely on human engineering of in-context (program) examples (ICEs) in the prompt. Moreover, ICEs are dataset- and task-specific. To generate them, significant labor by highly skilled programmers is required. For this reason, we argue that these methods *should not be called zero-shot* in their current form.

In this work, we present a framework that mitigates these issues, makes LLMs-as-programmers setup more robust, and removes the need for human engineering of ICEs. Our framework, whose effectiveness we show across a number of compositional question-answering and video temporal reasoning tasks with ViperGPT (but is universally applicable to other approaches), consists of the following:

- Firstly, instead of using a simple API with only basic routines that call individual tools, we introduce an "Abstract API". Abstract API consists of spatially and temporally abstract routines that remove the large burden on the LLM to have strong spatial and temporal reasoning.

- Second, instead of relying on a large number of dataset-specific (question, code)-pairs as ICEs, we introduce a setup that generates ICEs automatically. Using a few labeled examples (that are significantly cheaper to obtain, e.g. via crowd-sourcing), we generate query-code examples in a *zero-shot* manner and use these as ICEs. This mitigates the need for human engineering of ICEs.

- Third, we show how LLMs as controllers for visual reasoning can (to some extent) perform "self-correction" through "self-debugging" and "self-tuning" without any ground truth labels. In "self-debugging", we generate new code when the previous fails to execute, either by providing the LLM previous query-code pair and execution error as feedback or from scratch. In "self-tuning", we show how the tool hyperparameters can be tuned automatically if the execution fails due to a module.

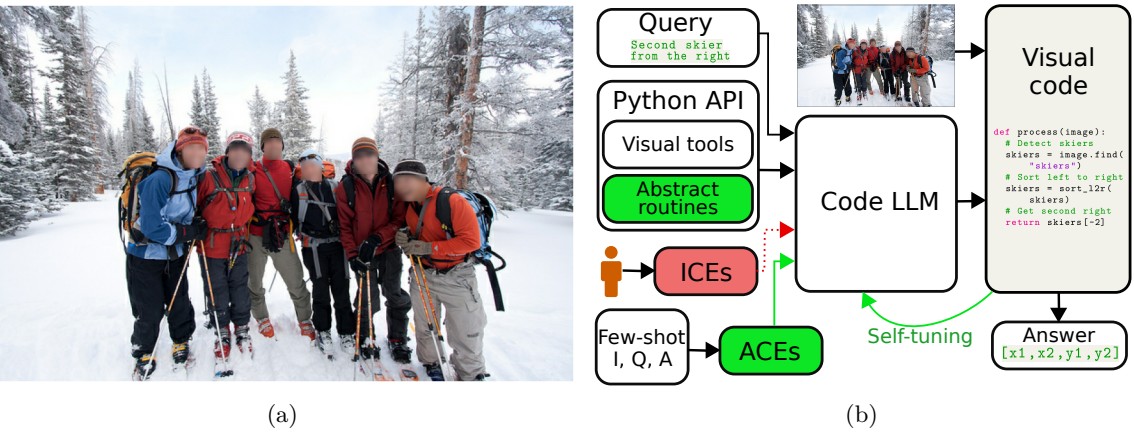

(a)  (b)

Figure 1: (a) RefCOCO (Yu et al., 2016) example image. (b) A code-generating LLM takes as input the query, the Python API (functions for "tool use" and Abstract API routines (functions) we introduce in Section 2.2) and a number of ICEs (we replace human-engineered ICEs by automatically-generated ACEs in Section 2.3). The LLM generates code that takes as input the image and outputs an answer (here a bounding box). If code fails to run, "self-tuning" (Section 2.4) can adjust parameters and generate new code.

## 2 LLMs as programmers for visual reasoning framework

In this section, we first provide a brief overview of the ViperGPT approach (Surís et al., 2023) on top of which we show the utility of our framework. We then describe each component of our framework, namely the Abstract API, automatic generation of ICEs, and self-correction.

### 2.1 Background

ViperGPT takes as input an image or a video and a textual query. The textual query is fed into an LLM (Codex (Chen et al., 2021)), together with the tools API and ICEs. The LLM generates a program that solves the given query using the tools without further training. The information in the prompt is crucial for good ViperGPT performance, as it is the only task-specific information provided. The prompt consists of a Python API with the necessary functions to solve the visual query, such as object detection, depth estimation, and language model queries. Additionally, ViperGPT uses several dataset-specific ICEs in the prompt. As we show in Section 3, performance depends heavily on these human-engineered examples.

ViperGPT API defines an `ImagePatch` and a `VideoSegment` class that contain image and video processing functions. Each function calls a pretrained model to compute the result. The API in the prompt does not contain function implementations, but it contains docstrings and query-code examples of their use. The ViperGPT API defines the following functions: `find` takes as input an image and a textual query, calls an open vocabulary detector and returns a list of image patches of detected objects; `exists` takes as input an image and a textual query and returns true if the query object exists in the image, otherwise false; `verify_property` takes as input an image, a noun representing an object and an attribute property to verify and returns a boolean whether the object has this property; `best_image_match` that takes as input a list of image patches and a textual query and returns the image patch that best matches the query; `best_text_match` that takes as input a list of queries and one image, and returns the query that best matches the image; `compute_depth` that computes the median depth of an image or image patch; `distance` which computes the pixel-distance between two images; `simple_query` which handles textual queries that are not decomposable, by calling an image captioning model; `select_answer` that given a context text describing a scene and a list of possible answers queries an LLM to select the correct answer. The `VideoSegment` class does not contain any functions that call individual models, but only the start and end point of the video segment and an iterator over the frames, which returns an `ImagePatch` object. For the full ViperGPT API, see Appendix A.4.

The code-generating LLM outputs code that attempts to solve the query. This code is executed, taking as input an image or a video (and optionally a list of possible answers) and outputs a result (e.g. a bounding box or a string). Due to generating programs in native Python code, ViperGPT avoids the need for custom interpreters and can leverage Python built-in functions (e.g. sort, if/else control flows, math functions, etc.).

## 2.2 Abstract API through visual routines

When programming, we continuously build new layers of abstraction. grouping them together into new functions. By abstracting away the implementation details, we reduce the cognitive load on the programmer which is then able to build systems of increased complexity. Motivated by this, we introduce a set of *spatially* and *temporally* abstract functions (routines [1]) that abstract away a number of lines of code for the same functionality and together make the *Abstract API*. From a practical perspective, we are motivated by a number of failure cases observed in the experiments (see Section 3). As presented in Section 2.1, ViperGPT's API is fairly simple (contains almost exclusively functions to call pretrained models). Although simplicity is good and often desirable, in the case of visual reasoning with LLMs as programmers, the lack of expressive visual routines requires the code-generating LLM to have strong spatial and temporal reasoning capabilities. Qualitative analysis showed that this is often not the case and that the current LLMs are not yet perfect in these terms (e.g. they confuse left and right, top and bottom). For example, for the query "return the second person from the right", the program generated by the LLM correctly sorts the persons along the horizontal axis but then wrongly takes the second index in the array (instead of the second last). Similarly, they sometimes "confuse" temporal order, e.g., if a "before" event means a smaller or a larger time index.

For these reasons, we introduce a set of spatially and temporally abstract routines. We add the following spatial routines: `get_patch_left_of`, `get_patch_right_of`, `get_patch_above_of`, `get_patch_below_of` for relational retrieval relative to a patch; `get_patch_closest_to_anchor_object` that sorts patches by their distance to an anchor object and returns the one with the smallest distance; `sort_patches_left_to_right`, `sort_patches_bottom_to_top`, and `sort_patches_front_to_back` to sort the list of patches along horizontal, vertical or depth axis; `get_middle_patch` to get the middle patch from a given list of image patches; For videos, we add temporal routines for event "localization": `get_video_segment_of_event`, `get_video_segment_before_event`, `get_video_segment_after_event`, and routines to either caption a video: `caption_video` or answer a simple question about the video: `simple_query`. The routines that we introduce are *general* in the sense that they are not specific to any individual task or dataset. This facilitates their reuse across tasks and avoids engineering task and dataset-specific routines. It is an open research question what the "optimal" set of primitive routines is. Another exciting research direction is using LLM with their own abstract routines, then reusing those to come up with even more abstract routines and so on. We leave these for future work.

## 2.3 Automatic generation of in-context examples

In-context examples (ICEs) greatly influence the performance of LLMs (Brown et al., 2020; Chen et al., 2023b). For example, ViperGPT (Surís et al., 2023) uses between 10 and 16 hand-engineered dataset-specific query-code ICEs per dataset. Similarly, VisProg (Gupta & Kembhavi, 2023) uses between 16 and 31 ICEs and CodeVQA (Subramanian et al., 2023) about 50 ICEs. However, constructing these ICEs requires heavy human engineering, as they might need to be rewritten in a way that the LLM can use them to "generalize" to other examples across the dataset. Furthermore, the constructed examples are specific not only to the dataset but also to the LLM and the API. If any of those changes, they need to be written from scratch. Finally, to write good query-code ICEs, highly skilled labor is required, ideally someone familiar with the workings of LLMs and a good grasp of Python programming.

In our work, we move beyond this need for human engineering of query-code ICEs. We start from a small set of labeled examples (e.g. 16 image-question-answer tuples), as is common in few-shot transfer learning (Zhai et al., 2019; Kolesnikov et al., 2020). We run our framework in a *zero-shot* manner (without any ICEs) on these few-shot examples, sort the results by accuracy, select the best-performing programs, pair them with the corresponding queries, and use them as ICEs during test time. We call such ICEs *automatically-generated*

---

[1]Note that our "routines" do not correspond to the visual routines of Ullman (1987) such as tracing or scanning.

*in-context examples* (ACEs). Importantly, *no gradient-based optimization is performed on the few-shot examples*. Intuitively, this works since even if the LLM does not always generate a program that solves the task correctly, it might sometimes come up with a correct program. Since retrieval is often easier than generating programs from scratch, the reuse of the correct programs improves performance on the test set.

ACEs provide a number of benefits over manually writing ICEs. First of all, ACEs are much cheaper to obtain as they do not require highly skilled labor to write them. Second, the algorithm that generates ACEs is general: it is neither specific to the API nor the LLM. If any of these changes, ACEs can be easily generated by re-running the algorithm. Furthermore, they can be seen as a first step of the LLM "coming up" with its own abstract rules and thus creating a "rulebook" (discussed in Section 2.2). Finally, few-shot (image, question, answer)-labeled examples are often available in datasets typically used in machine learning. If not available, they are cheap to obtain via crowd-sourcing and can be reused for further studies as a benchmark.

### 2.4 Self-correction

One of the advantages of solving visual reasoning tasks with LLMs as programmers is that we know when code fails to execute. The failure can happen, e.g. due to a compilation error (e.g. due to hallucination), some of the models failing, or a wrong return type (e.g. a bounding-box is expected, but code returns a string). Note that to detect these types of errors, no ground-truth labels are needed.

**Self-debugging.** If the code execution fails, we can query the code-generating LLM to correct the previously generated code. We do this by feeding back the query, the previously generated code, and the resulting error in the prompt (see the feedback template in Appendix A.3). Moreover, if the LLM's sampling temperature is higher than zero, we can query the model with a different random seed to generate new code from scratch. There are advantages to both of these approaches. If the code-generating LLM has good "self-correction" abilities, then it should be able to correct its own mistakes based on the feedback, as we humans could. However, if the LLM is not good at self-correction or does not know how to incorporate such feedback (e.g. if the LLM is not trained to be conversational), then feeding back the previous query and code will only "bias" the model to output the same solution. In that case, generating new code from scratch could work better.

**Self-tuning.** In some cases, we know that code execution failed due to some components of a particular module. For example, the open vocabulary detector fails due to a too high threshold hyperparameter. When the threshold is high, we have a higher number of false negatives. For such cases, we propose to automatically change the hyperparameter of the module (e.g. reduce the threshold) and execute code again. Although here we experiment with only a single option of tuning such hyperparameters, the idea is applicable to any model that involves such "sensitivity" hyperparameters. For example, another such candidate could be the threshold that determines the "similarity" of the CLIP-style image-text embedding model. The task of this module is to filter out a particular instance (e.g. "blue ball") from a set of instances ("balls"), e.g., based on an attribute ("blue"). It does so by comparing the text embedding of "blue" with the image embeddings of all "ball" image patches in the image. Then it returns all image patches whose embeddings have similarity with the text embedding higher than the threshold value. Here we would similarly tune the threshold in case this module fails.

## 3 Experiments

**Tasks.** We evaluate our method on four datasets: RefCOCO, RefCOCO+ (Yu et al., 2016), GQA (Hudson & Manning, 2019) and NExT-QA (Xiao et al., 2021) used in previous work (Surís et al., 2023). These datasets evaluate a diverse set of capabilities, namely visual grounding (RefCOCO, RefCOCO+), compositional image question answering (GQA), and video temporal reasoning (NExT-QA). In RefCOCO (example in Figure 1a), the task is to detect a bounding box of an object given its natural language description ("referring expression"). In compositional question answering in GQA, the task is to answer in natural language a compositional natural language query. We use the "test-dev" split of the GQA dataset, as in ViperGPT. In NExT-QA, the task is to answer a temporally compositional question by selecting one of the given multiple choice options. As in ViperGPT, we use NExT-QA "hard" split Buch et al. (2022). For RefCOCO and

Table 1: Comparison of our method against previous end-to-end and "LLMs as controllers" SotA methods. For "Ours (`code-bison`)", we report mean scores ± standard deviation across three random seeds. The reference numbers for SotA on each dataset are taken from the following publications: RefCOCO: ZS (Yang et al., 2023b), FS (Yao et al., 2021), Sup (Wang et al., 2022b); RefCOCO+: ZS (Yang et al., 2023b), FS (Yao et al., 2021), Sup (Wang et al., 2022b); GQA: ZS (Li et al., 2023b), FS (Jin et al., 2021), Sup (Nguyen et al., 2022); NExT-QA: ZS (Chen et al., 2023d), FS (Chen et al., 2023d), Sup (Ye et al., 2023a).

| Model | RefCOCO (IoU) | RefCOCO+ (IoU) | GQA (acc.) | NExT-QA (acc.) |
|---|---|---|---|---|
| Zero-Shot (ZS) SotA | 53.0 | 57.5 | 44.7 | 38.3 |
| Few-Shot (FS) SotA | 53.3 | 52.5 | 35.7 | 38.3 |
| Supervised (Sup) SotA | 94.0 | 91.7 | 72.1 | 63.1 |
| ViperGPT (paper) | 72.0 | 67.0 | 48.1 | 60.0 |
| ViperGPT (GitHub (GH) ZS) | 46.7 | - | - | - |
| ViperGPT (GH w/ DS-ICEs) | 60.5 | - | - | - |
| E2E bsl.(ZS OWLv2/PaLI-3) | 33.5 | 31.7 | 40.1 | 58.9 |
| E2E LLM-only baseline | - | - | - | 53.3 |
| Ours (`code-bison`, Zero-Shot) | 44.4 ± 0.9 | 38.2 ± 0.0 | 32.1 ± 0.4 | 60.2 ± 0.3 |
| Ours (`code-bison`) | 51.2 ± 0.2 | 45.7 ± 0.1 | 33.4 ± 0.2 | 61.0 ± 0.1 |

RefCOCO+, methods are evaluated in terms of Intersection over Union (IoU) between the detected and the ground truth bounding box and for GQA and NExT-QA in terms of accuracy of the predicted answer.

**Vision and Language Models.**   For code generation, we use a code instruction-tuned version of PaLM 2 (Anil et al., 2023) `code-bison` accessible via the Google Cloud API (Google, 2023). We use the same model to select an answer for the multichoice questions in the NExT-QA dataset. Vision models we use are OWLv2 (Minderer et al., 2023) for object detection, SigLiT (Zhai et al., 2023) for text-image comparison, MiDaS (Ranftl et al., 2020) for depth estimation, and PaLI-3 (Chen et al., 2023d) for image captioning and answering visual queries. Note that all models are different from the models that ViperGPT used (see Appendix A.2).

**Baselines.**   Strong baselines are essential for correctly measuring progress in machine learning. This is especially true in the emerging area of "tool use" (Cobbe et al., 2021; Komeili et al., 2021; Thoppilan et al., 2022; Parisi et al., 2022; Zeng et al., 2022; Gao et al., 2023a; Qin et al., 2023; Zhuge et al., 2023). When using an LLM and other pre-trained models, we must be careful to report the exact LLM version and/or API when it was accessed, and ideally report results over several random seeds to measure the statistical significance. In Table 1, we provide an overview of the previous Zero-Shot (ZS), Few-Shot (FS), and Supervised (Sup) SotA methods, ViperGPT, end-to-end (E2E) baselines, and our results on all datasets we used for evaluation.

Early in the project, we found it difficult to reproduce the results reported in the ViperGPT paper. Our first hypothesis is that this is due to differences in the vision and language models we use compared to the ViperGPT paper. However, when running the original ViperGPT code from the official GitHub repository on RefCOCO, we were only able to achieve an IoU of 60.5 as opposed to 72.0 reported in the paper. Note, however, that ViperGPT uses Codex, which is discontinued, so we use `GPT-3.5-turbo` (OpenAI, 2023). Also note that this score was obtained using 16 *dataset-specific* ICEs (DS-ICEs). These examples contain large amounts of dataset-specific human-engineered information, such as "clothing requires returning the person". In the case of truly Zero-Shot learning (without any human-engineered ICEs), the IoU score of ViperGPT's official GitHub code drops by 14 points to 46.7. Moreover, in their GitHub code we found hand-engineered improvements: if returning a bounding box fails, then return an "average" bounding box, and if code execution fails on GQA, then query the image captioner (BLIP-2 (Li et al., 2023b)). These code changes lead to improved results, but make it hard to quantify the true power of LLMs as controllers approach.

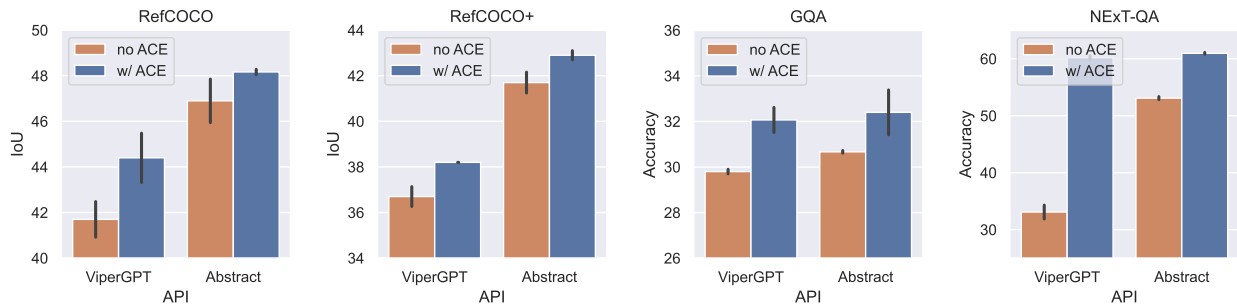

Figure 2: Using our Abstract API improves performance over the ViperGPT API across all datasets. Similarly, ACEs consistently improve performance, and these gains compound with the gains from the Abstract API. Uncertainty bars represent standard deviations computed over three random seeds.

In Table 1, we also provide Zero-Shot end-to-end baselines. On RefCOCO and RefCOCO+, we feed the query as input to the OWLv2 model and return its output. For the GQA end-to-end baseline, we query the PaLI-3 model (which was fine-tuned for multi-task inference on different captioning and VQA data sets, but not on GQA). For NExT-QA, we provide two baselines. For the first baseline, we subsample and caption with PaLI-3 one frame per second, and then feed all these captions to an LLM (`code-bison`) together with the question and multiple choice answers. As the second baseline, we simply feed the LLM the question and the possible answers and ask it to choose one. LLM-only baseline achieves an accuracy of 53.3%, which is only 5.4% lower than the baseline that also gets videos as input and significantly above the chance level accuracy of 20% (since there are 5 possible multichoice answers to each question). This suggests that NExT-QA might not fully evaluate the vision properties of the model and that new datasets are needed; for example, the Perception Test (Pătrăucean et al., 2023) was created specifically to avoid such problems.

Lastly, in Table 1 we report the Zero-Shot performance in our setup, as well as results for our best performing model variant (averaged over three random seeds). In the following sections, we evaluate each component, as well as their combinations, to obtain the reported results. Using `GPT-3.5-turbo` instead of `code-bison` resulted in a slight drop in performance, but as we shall see below, the same conclusions hold for both `code-bison` and `GPT-3.5-turbo` for all our suggested improvements. In the following, we present the components of our framework. For an ablation study, see Table 2 that shows that each component contributes positively to the final scores on each dataset.

### 3.1 Zero-Shot through spatially and temporally Abstract API

In Figure 2, we show the effect of using our Abstract API instead of the API used in ViperGPT. The API for RefCOCO, RefCOCO+, and GQA uses the same set of image routines (see Appendix A.4), whereas the API for NExT-QA uses only video-specific (temporal) routines. For now, we focus on the brown bars in Figure 2, and we compare the ViperGPT API and our Abstract API. We can see that our Abstract API leads to gains both with and without ACEs across all datasets. The performance gain is most notable for the NExT-QA dataset when ACEs are not used. We suspect that this is due to the LLM's difficulty in reasoning about the temporal order of events. This confirms our hypothesis that building a more abstract API such that the LLM does not need to use low-level Python routines is a promising direction.

Finally, we investigate whether our conclusions also hold for other LLMs, namely OpenAI's `GPT-3.5-turbo`. On RefCOCO `GPT-3.5-turbo` achieves an IoU of 28.9 with the ViperGPT API and 39.8 with our Abstract API and an accuracy of 9.4 and 42.9 for the ViperGPT API and our Abstract API, respectively, on NExT-QA. This confirms that our Abstract API brings gains not only for `code-bison`, but also for other LLMs. For each sample in the evaluation, we allow only one trial of code generation (no self-correction). Compared to the results with `code-bison`, IoU of 37.7 and 40.5 on RefCOCO and accuracy of 11.5 and 46.1 on NExT-QA for the ViperGPT API and our Abstract API, respectively, the results with `GPT-3.5-turbo` are slightly worse. We suspect that the reason for this could be that `GPT-3.5-turbo` is mainly built to be a conversational agent, while `code-bison` is specifically trained to have good coding capabilities.

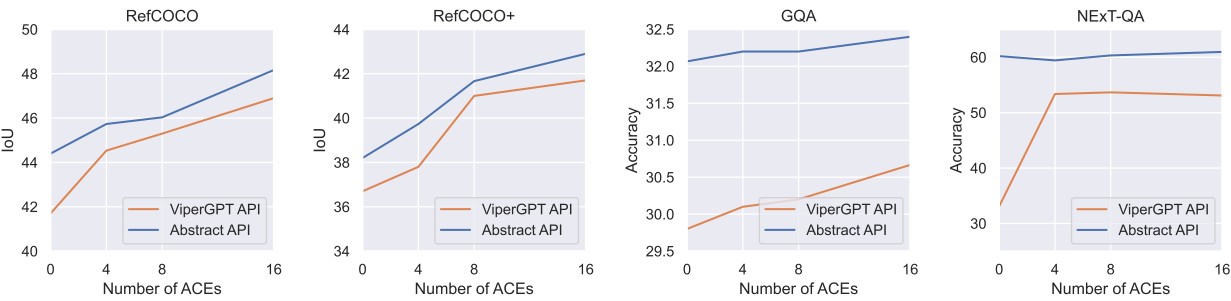

Figure 3: Increasing the number of ACEs in the prompt improves performance. Note that using the ViperGPT API on NExT-QA results in only three correct ACEs, so the performance plateaus after four ACEs.

### 3.2 Few-shot boostrapping via automatically generated in-context examples (ACEs)

We evaluate the effect of using automatically generated in-context examples (ACEs), described in Section 2.3. We can either sample few-shot examples manually or pick them at random. Both of these variants lead to good results, as we show in the following experiments. However, selecting examples manually allows for a better "quality" (in terms of diversity) given a small number of few-shot examples, so by default we use these for all experiments. For the first set of experiments, we manually pick 16 few-shot examples from the training set: image/video, question, and ground-truth answer. We try to make examples diverse to cover question "types" (e.g. left-right, front-back, closest-to, etc.).

In Figure 2, we show the effect of ACEs. For each dataset and for each API, the performance without ACEs is shown with the brown bar, and the performance with ACEs corresponds to the blue bar. We can see that for all datasets and all APIs, ACEs improve performance. The largest gains when using ACEs are for RefCOCO, RefCOCO+, and NExT-QA datasets when using the ViperGPT API. This indicates that ACEs are effective in dealing with complex spatial and temporal reasoning. More importantly, it can be seen in Figure 2 that the gains from both the Abstract API and ACEs compound for all tasks, indicating that they provide complementary strengths. Figure 3 shows how the performance in terms of IoU and accuracy scales with the number of few-shot examples used to generate ACEs. As expected, increasing the number of few-shot examples leads to improved performance. Note that the ViperGPT API on NExT-QA is able to correctly "solve" only 3 few-shot examples, so there are no gains beyond using 4 few-shot examples.

We evaluate the effect of using randomly sampled few-shot examples instead of manually selecting them. On RefCOCO, we sample 100 few-shot random samples from the training set, run Zero-Shot framework on them, sort the resulting programs by their IoU, and select the top 16 programs. Therefore, we end up with the same number of ACEs as with manual selection. On RefCOCO, we achieve IoU of 47.9 and 49.1 with the ViperGPT API and our Abstract API respectively. These results are slightly better than those with manually selected few-shot examples (46.9 and 48.2 IoU). This shows that the manual labor for generating ACEs can be removed altogether if we already have some labeled examples. With 50 few-shot random samples we obtain similar performance, and for 16 such samples we observe a small drop in performance (see Tables 5 and 6 in the Appendix A.1 for detailed results).

As in the previous section, we test whether our findings are consistent with `GPT-3.5-turbo` on RefCOCO and NExT-QA. On RefCOCO, when using `GPT-3.5-turbo`, ACEs improve IoU from 28.9 to 39.8 with the ViperGPT API and from 38.1 to 41.6 with our Abstract API. Similarly, for `GPT-3.5-turbo` on NExT-QA, ACEs improve accuracy from 9.4 to 42.9 the ViperGPT API and from 56.7 to 58.8 with our Abstract API. This confirms that the benefit of ACEs is not only limited to `code-bison` but also holds for `GPT-3.5-turbo` as well.

Another benefit of the few-shot setup when generating ACE is that it allows us to "tune" hyperparameters (HPs). For example, when sweeping over LLM temperature and object detection threshold HPs, we observed that the relative performances on the few-shot examples closely resemble the one when sweeping over the full validation dataset (the analysis is shown in Appendix A.1).

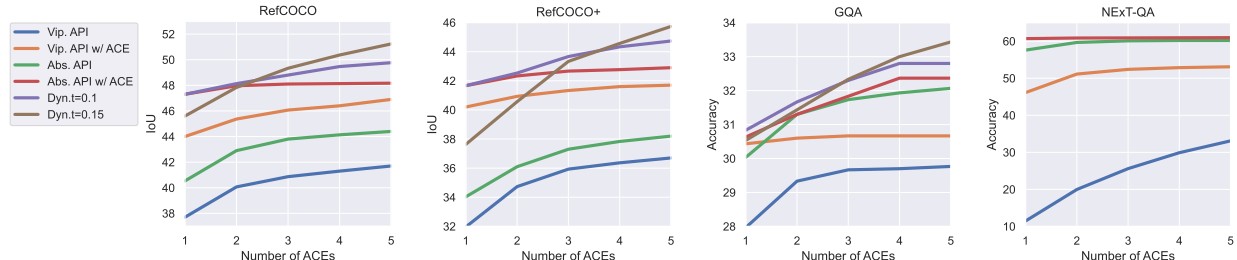

Figure 4: Increasing the number of "self-tuning" steps leads to improved performance. Our Abstract API (Abs. API) consistently outperforms the ViperGPT API (Vip. API). The best performance is achieved when using dynamic object detector threshold (Dyn.t) in addition to the Abstract API with ACE.

Table 2: Component-wise ablations of our framework. Each component contributes positively to the final score. Their relative contributions vary for different tasks. We report mean scores across three random seeds.

| Model | RefCOCO (IoU) | RefCOCO+ (IoU) | GQA (acc.) | NExT-QA (acc.) |
|---|---|---|---|---|
| ViperGPT API | 38.4 | 32.0 | 27.9 | 11.5 |
| + Abstract API | 42.3 (+3.9) | 34.0 (+2.0) | 30.0 (+2.1) | 57.6 (+46.1) |
| + ACE | 47.3 (+5.0) | 41.7 (+7.7) | 30.6 (+0.6) | 60.7 (+3.1) |
| + Self-debugging | 48.2 (+0.9) | 42.9 (+1.2) | 32.4 (+1.8) | **61.0** (+0.3) |
| + Self-tuning | **51.2** (+3.0) | **45.7** (+2.8) | **33.4** (+1.0) | - |

### 3.3 Self-correction

In this section, we analyze the ability of the framework to "self-correct" itself *without any external feedback* when code execution fails.

**Self-debugging.** When the program execution fails (due to e.g. compilation errors), we can retry by generating a new program (Chen et al., 2021). When creating a new program, we can also feed the previously generated code and the question as part of the prompt (see Appendix A.4), a variant that we call "self-debugging". Another option would be to simply repeat the exact same prompt as in the previous trial and rely on stochasticity in the LLM with a temperature greater than zero to generate a new correct solution. In our experiments, the "self-debugging" variant did not lead to an improvement in performance. In all cases, the performance plateaus after the first trial. This is in line with other recent findings Huang et al. (2023); Stechly et al. (2023); Valmeekam et al. (2023). On the other hand, the variant without any feedback in the prompt led to an increasingly better performance as the number of "trials" increases (see Figure 4).

**Self-tuning.** In some cases, we know that code fails due to some specific module. Therefore, we can then adjust the hyperparameters of the respective module and re-run code. For example, the open vocabulary detector we used (OWLv2) has a threshold hyperparameter that controls how sensitive it is. The lower this threshold, the more false positives we will have, but also the fewer false negatives. There is no global "optimal" value for this threshold that performs best for all images. Our framework allows us to adjust this hyperparameter dynamically: if the open vocabulary detector fails, we can lower the threshold and run the visual program again. In Figure 4, we can see that variants with a dynamic object detection threshold outperform all other variants and achieve the best performance. Note that the variant that achieves the highest performance after five trials has a lower performance for the first trial. This happens because we start with a higher object detection threshold value of 0.15 (by default, we use 0.1). In this case, initially there will be more false negatives, but also fewer false positives. As we decrease the threshold in subsequent trials, the previously false negatives are detected and the queries are correctly answered.

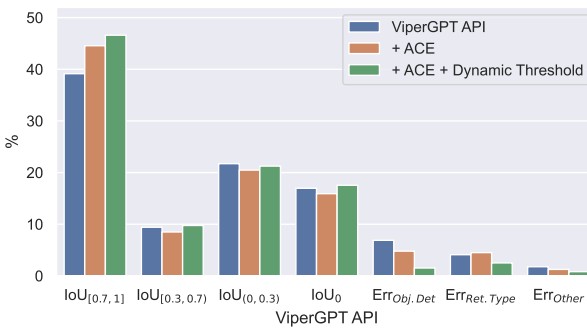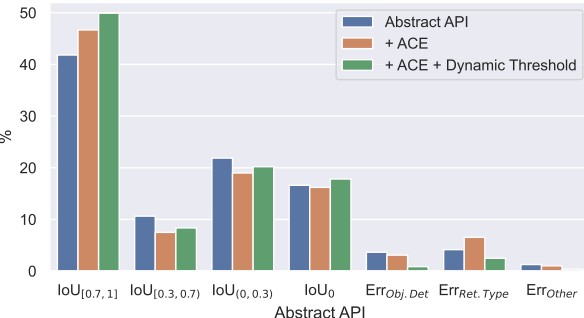

Figure 5: Error diagrams for the ViperGPT API and our Abstract API. We visualize the percentages of samples with IoU in certain ranges. "Err" classes are samples for which code execution failed due to either: object detection (Obj.Det), wrong return type (Ret.Type) or some other error (Other) e.g. hallucination.

### 3.4 Error analysis

Another benefit of visual reasoning with LLMs as programmers is interpretability. For example, we can get insights into the percentage of successful program executions, which can be further decomposed into the ones that resulted in correct or incorrect responses, and for the programs that failed to execute, we can provide further insights into why they failed, i.e. which module failed to return the correct result. Figure 5 shows one such error analysis on RefCOCO. Categories labeled with "Error" are the ones for which code failed to execute due to either object detection (Obj.Det), wrong return type (Ret.Type) or some other error (Other) e.g. hallucination. For all other cases, code executed correctly (it returned a bounding box), but sometimes it failed to detect the object ("IoU = 0" case). First, we notice that for both APIs the number of "correct" detections (IoU higher than 0.7) grows as we include ACEs and "self-tuning" through the dynamic object detection threshold. We can also see that the percentages of samples with high IoU are always higher for our Abstract API compared to the ViperGPT API. Finally, note that the percentage of error cases drops from 12.8% to 4.8% for the ViperGPT API and from 9.1% to 3.8% for our Abstract API.

## 4 Related work

**Visual reasoning with end-to-end monolithic models.** Recently, SotA on VQA has been largely obtained by scaling end-to-end vision and language models (VLMs) in terms of their size, training data, and compute (Alayrac et al., 2022; Chen et al., 2022b; Yu et al., 2022; Wang et al., 2022a; Gan et al., 2022; Lu et al., 2022; Li et al., 2023b; Driess et al., 2023; Chen et al., 2023d;c). One of the earliest VLMs Flamingo (Alayrac et al., 2022) used a frozen pretrained language model of up to 70B parameters and a frozen pretrained image encoder with 435M parameters and trained only a "cross-attention" module that served as an interface between them. Since then, efforts have mainly gone into scaling both the image and the language models: GIT (Wang et al., 2022a) used a 300M language and scaled the image encoder to 4.8B parameters; PaLI (Chen et al., 2022b) scaled both components jointly, language model to 17B and image encoder to 4B parameters; PaLI-X (Chen et al., 2023d) continued this trend of scaling the total number of parameters to 55B by using an image encoder with 22B parameters; PaLM-E scaled the number of total parameters in the VLM to 562B by integrating the 540B PaLM (Chowdhery et al., 2022) and the 22B Vision Transformer (Dosovitskiy et al., 2020; Dehghani et al., 2023). On the other hand, BLIP-2 (Li et al., 2023b) achieved SotA performance on various tasks with a 12B VLM and PaLI-3 Chen et al. (2023d) introduced a significantly smaller VLM with 5B total parameters that achieves competitive performance with SotA models on various VLM benchmarks. These models are typically pretrained on large amounts of data and then fine-tuned for the best performance on the downstream tasks. In contrast to this, visual reasoning with LLMs as programmers (Gupta & Kembhavi, 2023; Surís et al., 2023; Subramanian et al., 2023) does not require any fine-tuning or gradient updates on the downstream tasks. Moreover, even the largest VLMs struggle on the tasks that require compositional reasoning, the ability to generalize, fine-grained spatial capabilities, and counting (Bugliarello et al., 2023; Hsieh et al., 2023; Tschannen et al., 2023). Further scaling makes

them even more data- and compute-hungry; therefore, it is unclear whether scaling alone can solve these tasks. Conversely, using LLMs as programmers enables task decomposition into subtasks and holds promise of strong generalization and compositional reasoning.

**Visual reasoning with Modular Networks.**  Neural Modular Networks (NMNs) (Andreas et al., 2016; Johnson et al., 2017; Hu et al., 2017) are an alternative approach to monolithic end-to-end networks and offer a potential route to (compositional) generalization. They are also typically trained end-to-end, using supervised or reinforcement learning. NMNs are designed to have several modules, and the hope is that during training, each module will learn a different functionality, which will then be reusable across tasks. However, these models have a number of drawbacks: the program generation requires hand-tuned parsers, and they require optimization through reinforcement learning (e.g., REINFORCE Williams (1992)), which is often unstable. Learning all modules end-to-end hinders their ability to generalize (Bahdanau et al., 2018) and sometimes leads to a mode "collapse", where some modules take over all the work and other modules are never activated, or modules that do not learn intended functionalities (Subramanian et al., 2020). Furthermore, they sometimes require supervision for program learning, which is difficult to obtain at scale. On the other hand, visual reasoning with LLMs as programmers mitigates many issues of NMNs: it does not require gradient-based training or finetuning, it is able to incorporate any modules or swap the existing ones, it leverages the strong generalization ability of LLMs, and it generates programs by utilizing the in-context learning ability of LLMs, thereby removing the need for training program generators. The programs generated by LLM do not have to be domain specific, but can use a common language such as Python that does not require custom interpreters.

**Visual reasoning with LLMs as programmers.**  The field of LLMs as controllers for visual reasoning has received a great deal of interest recently. LLMs as controllers (also known as "tool use") approach became prominent in the literature (Parisi et al., 2022; Schick et al., 2023), in particular for structured reasoning in the natural language domain (Madaan et al., 2022; Wang et al., 2023b; Gao et al., 2023a; Chen et al., 2022a). In the domain of using LLMs as controllers for visual reasoning, PICa (Yang et al., 2022) solves a knowledge-based VQA task by first extracting an object and captions from the image and then querying GPT-3 with this information and in-context examples to answer a question. Socratic models (Zeng et al., 2022), HuggingGPT (Shen et al., 2023), Societies of Mind (Zhuge et al., 2023), and many other papers (Liu et al., 2023; Zhu et al., 2023b; Zhang et al., 2023b; Li et al., 2023c; Yang et al., 2024; Xu et al., 2022b; Zhang et al., 2023a; Gao et al., 2023b; Ye et al., 2023b; Yang et al., 2023c; Lu et al., 2024; Wu et al., 2023; Zhu et al., 2023a) (for a comprehensive survey, see Yin et al. (2023) or Li et al. (2023a)) compose vision and language models to "communicate" in a fixed "protocol" and solve tasks such as image captioning, visual question answering, image generation, and robot planning. On the other hand, models such as VisProg (Gupta & Kembhavi, 2023), ViperGPT (Surís et al., 2023) and CodeVQA (Subramanian et al., 2023) go beyond a fixed communication "protocol" by having LLM write (Python) programs. During execution, the program calls individual vision modules (such as the object detector and depth estimator) through an *API* that is provided in the prompt. Additionally, VisProg is also capable of generating images as program output. These models showed great performance and achieved SotA on tasks such as compositional visual question answering, visual grounding, and video temporal reasoning. However, in their current form, these models rely on heavy human engineering of query-code examples in the prompt that are dataset- and task-specific and require significant labor by highly skilled workers. Our framework, on the other hand, is able to automatically generate in-context examples, removing the need for humans to write query-code examples, uses an Abstract API that puts less burden on the LLM to have strong spatial and temporal reasoning abilities, and proposes a way to "self-correct" the programs that failed to execute. Note that our framework is equally applicable to all of the above approaches.

**Automatizing prompt engineering.**  Vast literature shows that prompt format and contents are often important for achieving good performance with an LLM (Reynolds & McDonell, 2021; Zhao et al., 2021; Lu et al., 2021; Moradi & Samwald, 2021; Madaan & Yazdanbakhsh, 2022; Wei et al., 2023). A prompt typically consists of a task description (in natural language), in-context examples (e.g. query-code in ViperGPT) and an API (in the case where LLMs write a program). Various prompting techniques have been engineered, such as Chain-of-Thought prompting (Wei et al., 2022), Self-Consistency (Wang et al., 2022c), Tree of Thoughts

(Yao et al., 2023), Graph of Thoughts (Besta et al., 2023), Plan-and-Solve Prompting (Wang et al., 2023a), Least-to-Most Prompting (Zhou et al., 2022a), etc. All these techniques rely on human prompt engineering, in particular, on in-context examples. On the other hand, some methods try to automate prompt engineering. They sometimes use gradient-based optimization (Shin et al., 2020; Gao et al., 2020; Wen et al., 2023) and some approaches require only API access to the model (Xu et al., 2022a; Prasad et al., 2022). Other works use LLMs for prompt optimization. APE (Zhou et al., 2022b) first generates instructions with an LLM, then selects instructions with the highest accuracy, and uses them for future LLM prompts. APO (Pryzant et al., 2023) generates feedback with an LLM that informs how to update the previous instruction. OPRO (Yang et al., 2023a) uses an LLM to generate new instructions at each optimization step, asking the LLM to improve task accuracy by changing task instructions, which requires determining the score on a small set of labeled examples and providing it in the meta-prompt. Promptbreeder (Fernando et al., 2023) goes a step further and proposes a self-referential self-improvement LLM using a meta-prompt that controls the generation of the main (task) prompt and evolves both via mutation. More importantly, Promptbreeder shows some surprising results such that a simple prompt "SOLUTION" outperforms all previous approaches. This further demonstrates the sensitivity of LLMs and the importance of automatizing the prompt engineering process. Common to all above frameworks for automatizing prompting is that they automatize the "task description" part of the prompt. On the other hand, in our framework, we automatize the generation of in-context examples, which might have an even greater influence on the performance of the LLM.

**LLMs and self-correction.** In the LLM literature, there have been mixed findings on the ability of LLMs to critique and self-correct their own reasoning and outputs. Self-Refine (Madaan et al., 2023) provides feedback to the LLM of the previously generated output, which is then refined. Several other approaches show benefits of providing feedback to LLM in improving reasoning (Shinn et al., 2024; Madaan et al., 2023), code generation (Chen et al., 2023e; Olausson et al., 2023; Chen et al., 2023a), improving LLM alignment (Bai et al., 2022; Ganguli et al., 2023), etc. On the other hand, there has been increasing evidence that LLMs cannot self-correct reasoning yet (Huang et al., 2023; Stechly et al., 2023; Valmeekam et al., 2023), unless they receive external feedback, which usually requires access to ground truth labels. In our work, we also found that providing the previous question and code as feedback to the model did not improve the results. However, we show that it is possible to improve performance by tuning hyperparameters on the fly, a direction that, to the best of our knowledge, has not been explored previously.

## 5 Discussion and future work

Although the LLMs as controllers framework is very promising for visual reasoning, there is much future work to be explored. First, the use of video-specific models (or tools) could greatly improve performance on video tasks compared to the image-specific models we used. Moreover, the code generating LLM currently only takes the question as the input, but for some questions the program that correctly solves the question can only be generated given the image or video as the input too.

The results with the Abstract API show that this is a promising path forward, but more research is needed to find the "optimal" set of visual and temporal routines. Starting from these primitive routines, the model should be able to build an ever-growing library of routines (e.g. through the ACE generation process) that it can later reuse. This growing library of routines will most likely grow larger than the size of the context window, so research is needed on an API "router" that can select routines that are relevant to a specific task at hand. Furthermore, it would be important to research ways of eliminating the need for few-shot examples when generating ACEs, e.g. by providing a natural language dataset specification (a datasheet).

Lastly, more effort should be put into creating better benchmarks for evaluating compositional visual reasoning, as current ones have a number of limitations. For example, not all samples in RefCOCO and RefCOCO+ require compositional reasoning, so the LLM should only query the open-vocabulary object detector. Similarly, many referring expressions in GQA are ambiguous in the sense that there is not a single unique answer. Finally, NExT-QA contains ambiguous questions (e.g. why someone did certain actions) or questions that can be answered by looking at the multiple choice answers only and disregarding the visual input altogether. The Perception Test (Pătrăucean et al., 2023) is a promising benchmark for future work, as it was specifically

created to avoid such problems. We hope that our findings inform future research on LLMs as controllers for visual reasoning and encourage systematic evaluation and benchmarking efforts in the future.

## 6 Conclusion

In this work, we present a framework that makes LLMs as programmers for visual reasoning more robust, removes the need for human engineering of in-context examples (ICEs), and thus brings them a step closer to *truly* zero-shot visual reasoners. We introduce an "Abstract API" that consists of spatially and temporally abstract routines, which improves performance by reducing the burden on the code-generating LLM to have strong spatial and temporal reasoning. By using a few labeled examples, we show how one can generate query-code ICEs automatically (ACEs) in a zero-shot manner. When used as in-context examples, ACEs consistently improve performance, eliminating the need for human engineering of ICEs. We demonstrate how LLMs as controllers for visual reasoning can (to a certain extent) perform "self-correction" through "self-debugging" and "self-tuning" without any ground-truth labels. In self-debugging, generating new code from scratch led to consistently better results, but providing the previous query-code pair as feedback to the LLM did not improve performance. In self-tuning, we show that the object detector threshold hyperparameter can be tuned automatically if code execution fails due to this module. Across a number of compositional question-answering and video temporal reasoning tasks, we demonstrate that each component of our framework consistently leads to improvement.

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

# A    Appendix

## A.1    Ablations

Here we present hyperparameter ablations, namely over the code-generating LLM (`code-bison`) temperature and the open-vocabulary object detector (OWLv2) threshold.

Table 3: Code-generating LLM (`code-bison`) temperature hyperparameter ablations (with ACEs). The scores are reported as mean ± standard deviation across three random seeds.

| Model | LLM temp. | RefCOCO (IoU) | RefCOCO+ (IoU) | NExT-QA (acc.) |
|---|---|---|---|---|
| ViperGPT API | 0.0 | 41.4 ± 0.3 | 39.8 ± 0.0 | 36.0 ± 0.1 |
| | 0.4 | 46.9 ± 0.8 | 41.7 ± 0.4 | 53.1 ± 0.1 |
| | 0.8 | 46.9 ± 0.3 | 42.0 ± 0.7 | 53.2 ± 0.1 |
| | 1.0 | 46.2 ± 0.7 | 40.9 ± 0.6 | 50.3 ± 0.5 |
| Abstract API | 0.0 | 47.0 ± 0.1 | 41.9 ± 0.3 | 59.1 ± 0.0 |
| | 0.4 | 48.2 ± 0.1 | **44.7** ± 0.2 | **61.0** ± 0.4 |
| | 0.8 | 48.2 ± 0.0 | 43.0 ± 0.2 | 60.6 ± 0.6 |
| | 1.0 | **48.8** ± 0.1 | 42.8 ± 0.3 | 59.9 ± 0.7 |

In Table 3, we report the scores for different `code-bison` LLM temperatures: 0, 0.4, 0.8 and 1.0. We found the deterministic case to underperform compared to the cases with a temperature higher than zero. This indicates that the solutions of which the models are most "confident" of are not necessarily always correct. On the other hand, when the temperature is too high, the model starts to hallucinate functions that do not exist in the API and the performance degrades. Early in our work, we settled on the `code-bison` LLM temperature of 0.4 and did not tune it further.

Table 4: Open-vocabulary object detector (OWLv2) threshold hyperparameter ablations. The scores are reported as mean ± standard deviation across three random seeds.

| Model | OWLv2 thrs. | RefCOCO (IoU) | RefCOCO+ (IoU) |
|---|---|---|---|
| ViperGPT API | 0.05 | 42.6 ± 0.3 | 41.9 ± 0.3 |
| | 0.10 | 46.9 ± 0.8 | 41.7 ± 0.4 |
| | 0.15 | 45.0 ± 0.5 | 38.2 ± 0.2 |
| | 0.20 | 33.3 ± 0.6 | 29.8 ± 0.2 |
| Abstract API | 0.05 | 42.8 ± 0.3 | 42.7 ± 0.7 |
| | 0.10 | **48.2** ± 0.1 | **44.7** ± 0.2 |
| | 0.15 | 47.1 ± 0.2 | 38.6 ± 0.1 |
| | 0.20 | 33.7 ± 0.4 | 30.1 ± 0.4 |

Table 4 shows the effect of using different thresholds for OWLv2 open vocabulary detector. This threshold controls the level of 'sensitivity' of the open vocabulary detector. If the threshold value is set too high, we will have fewer false positives, but also more false negatives. We perform this study on RefCOCO and RefCOCO+. On both datasets, the threshold of 0.1 achieves the best results, so by default we use this threshold in all our experiments.

In Table 5 and Table 6 we show results for RefCOCO and RefCOCO+ when using randomly sampled samples for the generation of ACEs. For comparison purposes, in the tables we also provide scores when not using any ACEs and with the default setting when generating ACEs with 16 manually selected few-shot examples. From the tables, we can see that randomly selected 100 or 50 samples perform similarly as when using 16 manual samples. With 16 random samples we observe a small drop in performance, though we still observe

Table 5: Results on RefCOCO with randomly sampled few-shot samples for generating ACEs. Shown are IoU scores for zero-shot (w/o ACE) setting, with the default setting that we used throughout the paper (16 manually sampled few-shot samples from the dataset (16 manual)) and with 100, 50 and 16 randomly-sampled samples (100, 50 and 16 random) from the dataset. The IoU scores are reported as mean $\pm$ standard deviation across three random seeds.

| API | w/o ACE | 16 manual | 100 random | 50 random | 16 random |
|---|---|---|---|---|---|
| ViperGPT | $41.7 \pm 0.6$ | $46.9 \pm 0.8$ | $47.9 \pm 0.2$ | $45.8 \pm 0.4$ | $41.5 \pm 0.2$ |
| Abstract | $44.4 \pm 0.9$ | $48.2 \pm 0.1$ | $49.1 \pm 0.2$ | $49.0 \pm 0.2$ | $46.9 \pm 0.2$ |

Table 6: Results on RefCOCO+ with randomly sampled few-shot samples for generating ACEs. Shown are IoU scores for zero-shot (w/o ACE) setting, with the default setting that we used throughout the paper (16 manually sampled few-shot samples from the dataset (16 manual)) and with 100, 50 and 16 randomly-sampled samples (100, 50 and 16 random) from the dataset. The IoU scores are reported as mean $\pm$ standard deviation across three random seeds.

| API | w/o ACE | 16 manual | 100 random | 50 random | 16 random |
|---|---|---|---|---|---|
| ViperGPT | $36.7 \pm 0.4$ | $41.7 \pm 0.4$ | $41.3 \pm 0.1$ | $40.3 \pm 0.3$ | $36.3 \pm 0.1$ |
| Abstract | $38.2 \pm 0.0$ | $42.9 \pm 0.2$ | $42.8 \pm 0.6$ | $41.8 \pm 0.0$ | $40.2 \pm 0.4$ |

an improvement compared to the setting without any ACEs. In summary, this shows that the manual labor for generating ACEs can be removed altogether if we already have some labeled examples.

### A.2 Pretrained models

Here we specify the pretrained models we used, and compare them with the ones used in ViperGPT:

- Open-vocabulary object detector:
  - Ours: OWLv2 (Minderer et al., 2023).
  - ViperGPT: GLIP Li et al. (2022) from the official GitHub repository[2].

- Depth estimation model:
  - Ours: MiDaS (Ranftl et al., 2020) v2 "DPT_Small" from PyTorch hub[3].
  - ViperGPT: MiDaS (Ranftl et al., 2020) v2 "DPT_Large" version from the PyTorch hub[4].

- Vision-language captioning model:
  - Ours: PaLI-3 (Chen et al., 2023d).
  - ViperGPT: BLIP-2 (Li et al., 2023b) from the official repository[5].

- CLIP-style image-text embedding model:
  - Ours: SigLiT (Zhai et al., 2023).
  - ViperGPT: X-VLM (Zeng et al., 2021) version finetuned for retrieval on MSCOCO from the official repository[6].

- Code-generating LLM:
  - Ours: `code-bison` accessible via the Google Cloud Vertex AI API (Google, 2023).
  - ViperGPT: Codex (`code-davinci-002`) via the official OpenAI Python API[7].

- Answer selector (based on context information) LLM for multiple choice questions in NExT-QA:
  - Ours: `code-bison` accessible via the Google Cloud Vertex AI API (Google, 2023).
  - ViperGPT: GPT-3 via the official OpenAI Python API[8].

### A.3 Self-debugging prompt

```
prompt += f"""

Previously, for the query:
# {query}
you've generated the following code:
{code}

The execution of the above code failed and returned the following error message:
{error}.

Given this, correct the above function, such that it executes correctly and solves the same
    query.

"""
```

---

[2]https://github.com/microsoft/GLIP
[3]https://pytorch.org/hub/intelisl_midas_v2/
[4]https://pytorch.org/hub/intelisl_midas_v2/
[5]https://github.com/salesforce/LAVIS/tree/main/projects/blip2
[6]https://github.com/zengyan-97/X-VLM
[7]https://openai.com/blog/openai-api
[8]https://openai.com/blog/openai-api

## A.4  Prompt listings

### A.4.1  RefCOCO and GQA - ViperGPT API

```python
import math

class ImagePatch:
    """A Python class containing a crop of an image centered around a particular object, as
    well as relevant information.
    Attributes
    ----------
    cropped_image : array_like
        An array-like of the cropped image taken from the original image.
    left, lower, right, upper : int
        An int describing the position of the (left/lower/right/upper) border of the crop's
    bounding box in the original image.

    Methods
    -------
    find(object_name: str)->List[ImagePatch]
        Returns a list of new ImagePatch objects containing crops of the image centered
    around any objects found in the
        image matching the object_name.
    exists(object_name: str)->bool
        Returns True if the object specified by object_name is found in the image, and False
     otherwise.
    verify_property(property: str)->bool
        Returns True if the property is met, and False otherwise.
    compute_depth()->float
        Returns the median depth of the image crop.
    crop(left: int, lower: int, right: int, upper: int)->ImagePatch
        Returns a new ImagePatch object containing a crop of the image at the given
    coordinates.
    """

    def __init__(self, image, left: int = None, lower: int = None, right: int = None, upper:
     int = None):
        """Initializes an ImagePatch object by cropping the image at the given coordinates
    and stores the coordinates as
        attributes. If no coordinates are provided, the image is left unmodified, and the
    coordinates are set to the
        dimensions of the image.
        Parameters
        -------
        image : array_like
            An array-like of the original image.
        left, lower, right, upper : int
            An int describing the position of the (left/lower/right/upper) border of the
    crop's bounding box in the original image.
        """
        if left is None and right is None and upper is None and lower is None:
            self.cropped_image = image
            self.left = 0
            self.lower = 0
            self.right = image.shape[2]  # width
            self.upper = image.shape[1]  # height
        else:
            self.cropped_image = image[:, lower:upper, left:right]
            self.left = left
            self.upper = upper
            self.right = right
            self.lower = lower

        self.width = self.cropped_image.shape[2]
        self.height = self.cropped_image.shape[1]

        self.horizontal_center = (self.left + self.right) / 2
        self.vertical_center = (self.lower + self.upper) / 2
```

```
56
57     def find(self, object_name: str) -> List[ImagePatch]:
58         """Returns a list of ImagePatch objects matching object_name contained in the crop
       if any are found.
59         Otherwise, returns an empty list.
60         Parameters
61         ----------
62         object_name : str
63             the name of the object to be found
64
65         Returns
66         -------
67         List[ImagePatch]
68             a list of ImagePatch objects matching object_name contained in the crop
69
70         Examples
71         --------
72         >>> # return the foo
73         >>> def execute_command(image) -> List[ImagePatch]:
74         >>>     image_patch = ImagePatch(image)
75         >>>     foo_patches = image_patch.find("foo")
76         >>>     return foo_patches
77         """
78         return find_in_image(self.cropped_image, object_name)
79
80     def exists(self, object_name: str) -> bool:
81         """Returns True if the object specified by object_name is found in the image, and
       False otherwise.
82         Parameters
83         -------
84         object_name : str
85             A string describing the name of the object to be found in the image.
86
87         Examples
88         -------
89         >>> # Are there both foos and garply bars in the photo?
90         >>> def execute_command(image)->str:
91         >>>     image_patch = ImagePatch(image)
92         >>>     is_foo = image_patch.exists("foo")
93         >>>     is_garply_bar = image_patch.exists("garply bar")
94         >>>     return is_foo and is_garply_bar
95         """
96         return len(self.find(object_name)) > 0
97
98     def verify_property(self, object_name: str, visual_property: str) -> bool:
99         """Returns True if the object possesses the visual property, and False otherwise.
100        Differs from 'exists' in that it presupposes the existence of the object specified
       by object_name, instead checking whether the object possesses the property.
101        Parameters
102        -------
103        object_name : str
104            A string describing the name of the object to be found in the image.
105        visual_property : str
106            A string describing the simple visual property (e.g., color, shape, material) to
        be checked.
107
108        Examples
109        -------
110        >>> # Do the letters have blue color?
111        >>> def execute_command(image) -> str:
112        >>>     image_patch = ImagePatch(image)
113        >>>     letters_patches = image_patch.find("letters")
114        >>>     # Question assumes only one letter patch
115        >>>     return letters_patches[0].verify_property("letters", "blue")
116        """
117        return verify_property(self.cropped_image, object_name, property)
118
119    def compute_depth(self):
```

```
120          """Returns the median depth of the image crop
121          Parameters
122          ----------
123          Returns
124          -------
125          float
126              the median depth of the image crop
127
128          Examples
129          --------
130          >>> # the bar furthest away
131          >>> def execute_command(image)->ImagePatch:
132          >>>     image_patch = ImagePatch(image)
133          >>>     bar_patches = image_patch.find("bar")
134          >>>     bar_patches.sort(key=lambda bar: bar.compute_depth())
135          >>>     return bar_patches[-1]
136          """
137          depth_map = compute_depth(self.cropped_image)
138          return depth_map.median()
139
140      def crop(self, left: int, lower: int, right: int, upper: int) -> ImagePatch:
141          """Returns a new ImagePatch cropped from the current ImagePatch.
142          Parameters
143          -------
144          left, lower, right, upper : int
145              The (left/lower/right/upper)most pixel of the cropped image.
146          -------
147          """
148          return ImagePatch(self.cropped_image, left, lower, right, upper)
149
150      def overlaps_with(self, left, lower, right, upper):
151          """Returns True if a crop with the given coordinates overlaps with this one,
152          else False.
153          Parameters
154          ----------
155          left, lower, right, upper : int
156              the (left/lower/right/upper) border of the crop to be checked
157
158          Returns
159          -------
160          bool
161              True if a crop with the given coordinates overlaps with this one, else False
162
163          Examples
164          --------
165          >>> # black foo on top of the qux
166          >>> def execute_command(image) -> ImagePatch:
167          >>>     image_patch = ImagePatch(image)
168          >>>     qux_patches = image_patch.find("qux")
169          >>>     qux_patch = qux_patches[0]
170          >>>     foo_patches = image_patch.find("black foo")
171          >>>     for foo in foo_patches:
172          >>>         if foo.vertical_center > qux_patch.vertical_center
173          >>>             return foo
174          """
175          return self.left <= right and self.right >= left and self.lower <= upper and self.
     upper >= lower
176
177
178 def best_image_match(list_patches: List[ImagePatch], content: List[str], return_index=False)
         -> Union[ImagePatch, int]:
179      """Returns the patch most likely to contain the content.
180      Parameters
181      ----------
182      list_patches : List[ImagePatch]
183      content : List[str]
184          the object of interest
185      return_index : bool
```

```
186            if True, returns the index of the patch most likely to contain the object
187
188      Returns
189      -------
190      int
191          Patch most likely to contain the object
192      """
193      return best_image_match(list_patches, content, return_index)
194
195
196 def distance(patch_a: ImagePatch, patch_b: ImagePatch) -> float:
197      """
198      Returns the distance between the edges of two ImagePatches. If the patches overlap, it
         returns a negative distance
199      corresponding to the negative intersection over union.
200
201      Parameters
202      ----------
203      patch_a : ImagePatch
204      patch_b : ImagePatch
205
206      Examples
207      --------
208      # Return the qux that is closest to the foo
209      >>> def execute_command(image):
210      >>>     image_patch = ImagePatch(image)
211      >>>     qux_patches = image_patch.find('qux')
212      >>>     foo_patches = image_patch.find('foo')
213      >>>     foo_patch = foo_patches[0]
214      >>>     qux_patches.sort(key=lambda x: distance(x, foo_patch))
215      >>>     return qux_patches[0]
216      """
217      return distance(patch_a, patch_b)
218
219 INSERT_IN_CONTEXT_EXAMPLES_HERE
220
221 Write a function using Python and the ImagePatch class (above) that could be executed to
        provide an answer to the query.
222
223 Consider the following guidelines:
224 - Use base Python (comparison, sorting) for basic logical operations, left/right/up/down,
        math, etc.
225 - Make sure to always return an ImagePatch object.
226 - Make sure that for all possible control flows, the program always returns an ImagePatch
        object.
227
228 INSERT_PREVIOUS_CODE_AND_ERROR_HERE
229
230 # INSERT_QUERY_HERE
```

### A.4.2 RefCOCO and GQA - Abstract API

```python
import math

class ImagePatch:
    """A Python class containing a crop of an image centered around a particular object, as
    well as relevant information.
    Attributes
    ----------
    cropped_image : array_like
        An array-like of the cropped image taken from the original image.
    left, lower, right, upper : int
        An int describing the position of the (left/lower/right/upper) border of the crop's
    bounding box in the original image.

    Methods
    -------
    find(object_name: str)->List[ImagePatch]
        Returns a list of new ImagePatch objects containing crops of the image centered
    around any objects found in the
        image matching the object_name.
    exists(object_name: str)->bool
        Returns True if the object specified by object_name is found in the image, and False
     otherwise.
    verify_property(property: str)->bool
        Returns True if the property is met, and False otherwise.
    compute_depth()->float
        Returns the median depth of the image crop.
    crop(left: int, lower: int, right: int, upper: int)->ImagePatch
        Returns a new ImagePatch object containing a crop of the image at the given
    coordinates.
    """

    def __init__(self, image, left: int = None, lower: int = None, right: int = None, upper:
     int = None):
        """Initializes an ImagePatch object by cropping the image at the given coordinates
    and stores the coordinates as
        attributes. If no coordinates are provided, the image is left unmodified, and the
    coordinates are set to the
        dimensions of the image.
        Parameters
        -------
        image : array_like
            An array-like of the original image.
        left, lower, right, upper : int
            An int describing the position of the (left/lower/right/upper) border of the
    crop's bounding box in the original image.
        """
        if left is None and right is None and upper is None and lower is None:
            self.cropped_image = image
            self.left = 0
            self.lower = 0
            self.right = image.shape[2]  # width
            self.upper = image.shape[1]  # height
        else:
            self.cropped_image = image[:, lower:upper, left:right]
            self.left = left
            self.upper = upper
            self.right = right
            self.lower = lower

        self.width = self.cropped_image.shape[2]
        self.height = self.cropped_image.shape[1]

        self.horizontal_center = (self.left + self.right) / 2
        self.vertical_center = (self.lower + self.upper) / 2

    def find(self, object_name: str) -> List[ImagePatch]:
```

```
58          """Returns a list of ImagePatch objects matching object_name contained in the crop
     if any are found.
59          Otherwise, returns an empty list.
60          Parameters
61          ----------
62          object_name : str
63              the name of the object to be found
64
65          Returns
66          -------
67          List[ImagePatch]
68              a list of ImagePatch objects matching object_name contained in the crop
69
70          Examples
71          --------
72          >>> # return the foo
73          >>> def execute_command(image) -> List[ImagePatch]:
74          >>>     image_patch = ImagePatch(image)
75          >>>     foo_patches = image_patch.find("foo")
76          >>>     return foo_patches
77          """
78          return find_in_image(self.cropped_image, object_name)
79
80      def exists(self, object_name: str) -> bool:
81          """Returns True if the object specified by object_name is found in the image, and
     False otherwise.
82          Parameters
83          -------
84          object_name : str
85              A string describing the name of the object to be found in the image.
86
87          Examples
88          -------
89          >>> # Are there both foos and garply bars in the photo?
90          >>> def execute_command(image)->str:
91          >>>     image_patch = ImagePatch(image)
92          >>>     is_foo = image_patch.exists("foo")
93          >>>     is_garply_bar = image_patch.exists("garply bar")
94          >>>     return is_foo and is_garply_bar
95          """
96          return len(self.find(object_name)) > 0
97
98      def verify_property(self, object_name: str, visual_property: str) -> bool:
99          """Returns True if the object possesses the visual property, and False otherwise.
100         Differs from 'exists' in that it presupposes the existence of the object specified
     by object_name, instead checking whether the object possesses the property.
101         Parameters
102         -------
103         object_name : str
104             A string describing the name of the object to be found in the image.
105         visual_property : str
106             A string describing the simple visual property (e.g., color, shape, material) to
      be checked.
107
108         Examples
109         -------
110         >>> # Do the letters have blue color?
111         >>> def execute_command(image) -> str:
112         >>>     image_patch = ImagePatch(image)
113         >>>     letters_patches = image_patch.find("letters")
114         >>>     # Question assumes only one letter patch
115         >>>     return letters_patches[0].verify_property("letters", "blue")
116         """
117         return verify_property(self.cropped_image, object_name, property)
118
119     def compute_depth(self):
120         """Returns the median depth of the image crop
121         Parameters
```

```
122              ----------
123              Returns
124              -------
125              float
126                  the median depth of the image crop
127
128              Examples
129              --------
130              >>> # the bar furthest away
131              >>> def execute_command(image)->ImagePatch:
132              >>>     image_patch = ImagePatch(image)
133              >>>     bar_patches = image_patch.find("bar")
134              >>>     bar_patches.sort(key=lambda bar: bar.compute_depth())
135              >>>     return bar_patches[-1]
136              """
137              depth_map = compute_depth(self.cropped_image)
138              return depth_map.median()
139
140          def crop(self, left: int, lower: int, right: int, upper: int) -> ImagePatch:
141              """Returns a new ImagePatch cropped from the current ImagePatch.
142              Parameters
143              -------
144              left, lower, right, upper : int
145                  The (left/lower/right/upper)most pixel of the cropped image.
146              -------
147              """
148              return ImagePatch(self.cropped_image, left, lower, right, upper)
149
150          def overlaps_with(self, left, lower, right, upper):
151              """Returns True if a crop with the given coordinates overlaps with this one,
152              else False.
153              Parameters
154              ----------
155              left, lower, right, upper : int
156                  the (left/lower/right/upper) border of the crop to be checked
157
158              Returns
159              -------
160              bool
161                  True if a crop with the given coordinates overlaps with this one, else False
162
163              Examples
164              --------
165              >>> # black foo on top of the qux
166              >>> def execute_command(image) -> ImagePatch:
167              >>>     image_patch = ImagePatch(image)
168              >>>     qux_patches = image_patch.find("qux")
169              >>>     qux_patch = qux_patches[0]
170              >>>     foo_patches = image_patch.find("black foo")
171              >>>     for foo in foo_patches:
172              >>>         if foo.vertical_center > qux_patch.vertical_center
173              >>>             return foo
174              """
175              return self.left <= right and self.right >= left and self.lower <= upper and self.
          upper >= lower
176
177
178  def best_image_match(list_patches: List[ImagePatch], content: List[str], return_index=False)
          -> Union[ImagePatch, int]:
179      """Returns the patch most likely to contain the content.
180      Parameters
181      ----------
182      list_patches : List[ImagePatch]
183      content : List[str]
184          the object of interest
185      return_index : bool
186          if True, returns the index of the patch most likely to contain the object
187
```

```
188        Returns
189        -------
190        int
191            Patch most likely to contain the object
192        """
193        return best_image_match(list_patches, content, return_index)
194

195

196  def distance(patch_a: ImagePatch, patch_b: ImagePatch) -> float:
197        """
198        Returns the distance between the edges of two ImagePatches. If the patches overlap, it
      returns a negative distance
199        corresponding to the negative intersection over union.
200

201        Parameters
202        ----------
203        patch_a : ImagePatch
204        patch_b : ImagePatch
205

206        Examples
207        --------
208        # Return the qux that is closest to the foo
209        >>> def execute_command(image):
210        >>>     image_patch = ImagePatch(image)
211        >>>     qux_patches = image_patch.find('qux')
212        >>>     foo_patches = image_patch.find('foo')
213        >>>     foo_patch = foo_patches[0]
214        >>>     qux_patches.sort(key=lambda x: distance(x, foo_patch))
215        >>>     return qux_patches[0]
216        """
217        return distance(patch_a, patch_b)
218

219  def get_patch_left_of(patch: ImagePatch) -> ImagePatch:
220        left_patch = get_patch_left_of(patch)
221        return left_patch
222

223  def get_patch_right_of(patch: ImagePatch) -> ImagePatch:
224        right_patch = get_patch_right_of(patch)
225        return right_patch
226

227  def get_patch_above_of(patch: ImagePatch) -> ImagePatch:
228        above_patch = get_patch_above_of(patch)
229        return above_patch
230

231  def get_patch_below_of(patch: ImagePatch) -> ImagePatch:
232        below_patch = get_patch_below_of(patch)
233        return below_patch
234

235  def get_patch_around_of(patch: ImagePatch) -> ImagePatch:
236        around_patch = get_patch_around_of(patch)
237        return around_patch
238

239  def sort_patches_left_to_right(list_patches: List[ImagePatch]) -> List[ImagePatch]:
240        """
241        Sorts patches according to their horizontal centers.
242

243        Parameters
244        ----------
245        list_patches : List[ImagePatch]
246

247        Examples
248        --------
249        # Right foo
250        >>> def execute_command(image):
251        >>>     image_patch = ImagePatch(image)
252        >>>     foo_patches = image_patch.find('foo')
253        >>>     foo_patches = sort_patches_left_to_right(foo_patches)
254        >>>     right_foo_patch = foo_patches[-1]
```

```
255     >>>      return right_foo_patch
256     """
257     return sort_patches_left_to_right(list_patches)
258
259
260 def sort_patches_bottom_to_top(list_patches: List[ImagePatch]) -> List[ImagePatch]:
261     """
262     Sorts patches according to their vertical centers.
263
264     Parameters
265     ----------
266     list_patches : List[ImagePatch]
267
268     Examples
269     --------
270     # Second bar from the top
271     >>> def execute_command(image):
272     >>>     image_patch = ImagePatch(image)
273     >>>     bar_patches = image_patch.find('bar')
274     >>>     bar_patches = sort_patches_bottom_to_top(bar_patches)
275     >>>     second_topmost_bar_patch = bar_patches[-2]
276     >>>     return second_topmost_bar_patch
277     """
278     return sort_patches_bottom_to_top(list_patches)
279
280
281 def sort_patches_front_to_back(list_patches: List[ImagePatch]) -> List[ImagePatch]:
282     """
283     Sorts patches according to how far from camera they are.
284
285     Parameters
286     ----------
287     list_patches : List[ImagePatch]
288
289     Examples
290     --------
291     # Person in the back
292     >>> def execute_command(image):
293     >>>     image_patch = ImagePatch(image)
294     >>>     person_patches = image_patch.find('person')
295     >>>     person_patches = sort_patches_front_to_back(person_patches)
296     >>>     person_in_the_back = person_patches[-1]
297     >>>     return person_in_the_back
298     """
299     return sort_patches_front_to_back(list_patches)
300
301
302 def get_middle_patch(list_patches: List[ImagePatch]) -> ImagePatch:
303     """
304     Returns the middle patch.
305
306     Parameters
307     ----------
308     list_patches : List[ImagePatch]
309
310     Examples
311     --------
312     # Middle ham
313     >>> def execute_command(image):
314     >>>     image_patch = ImagePatch(image)
315     >>>     ham_patches = image_patch.find('ham')
316     >>>     middle_ham_patch = get_middle_patch(ham_patches)
317     >>>     return middle_ham_patch
318     """
319     return get_middle_patch(list_patches)
320
321
```

```
322  def get_patch_closest_to_anchor_object(list_patches: List[ImagePatch], anchor_object:
         ImagePatch) -> ImagePatch:
323      """
324      Returns the object from list_patches that is the closest to the anchor_object.
325
326      Parameters
327      ----------
328      list_patches : List[ImagePatch]
329      anchor_object : ImagePatch
330
331      Examples
332      --------
333      # Foo next to bar
334      >>> def execute_command(image):
335      >>>     image_patch = ImagePatch(image)
336      >>>     foo_patches = image_patch.find('foo')
337      >>>     bar_patches = image_patch.find('bar')
338      >>>     bar_patch = bar_patches[0]
339      >>>     foo_next_to_bar_patch = get_patch_closest_to_anchor_object(foo_patches,
         bar_patch)
340      >>>     return foo_next_to_bar_patch
341      """
342      return get_patch_closest_to_anchor_object(list_patches, anchor_object)
343
344
345  INSERT_IN_CONTEXT_EXAMPLES_HERE
346
347  Write a function using Python and the ImagePatch class (above) that could be executed to
         provide an answer to the query.
348
349  Consider the following guidelines:
350  - Use base Python (comparison, sorting) for basic logical operations, left/right/up/down,
         math, etc.
351  - Make sure to always return an ImagePatch object.
352  - Make sure that for all possible control flows, the program always returns an ImagePatch
         object.
353  - ImagePatch class uses left and right to denote horizontal edges.
354  - ImagePatch class uses bottom and top to denote vertical edges.
355
356  INSERT_PREVIOUS_CODE_AND_ERROR_HERE
357
358  # INSERT_QUERY_HERE
```

### A.4.3 NExT-QA - ViperGPT API

```python
import math

class ImagePatch:
    """A Python class containing a crop of an image centered around a particular object, as
    well as relevant information.
    Attributes
    ----------
    cropped_image : array_like
        An array-like of the cropped image taken from the original image.
    left, lower, right, upper : int
        An int describing the position of the (left/lower/right/upper) border of the crop's
    bounding box in the original image.

    Methods
    -------
    find(object_name: str)->List[ImagePatch]
        Returns a list of new ImagePatch objects containing crops of the image centered
    around any objects found in the
        image matching the object_name.
    exists(object_name: str)->bool
        Returns True if the object specified by object_name is found in the image, and False
     otherwise.
    verify_property(property: str)->bool
        Returns True if the property is met, and False otherwise.
    best_text_match(option_list: List[str], prefix: str)->str
        Returns the string that best matches the image.
    simple_query(question: str=None)->str
        Returns the answer to a basic question asked about the image. If no question is
    provided, returns the answer to "What is this?".
    compute_depth()->float
        Returns the median depth of the image crop.
    crop(left: int, lower: int, right: int, upper: int)->ImagePatch
        Returns a new ImagePatch object containing a crop of the image at the given
    coordinates.
    """

    def __init__(self, image, left: int = None, lower: int = None, right: int = None, upper:
     int = None):
        """Initializes an ImagePatch object by cropping the image at the given coordinates
    and stores the coordinates as
        attributes. If no coordinates are provided, the image is left unmodified, and the
    coordinates are set to the
        dimensions of the image.
        Parameters
        -------
        image : array_like
            An array-like of the original image.
        left, lower, right, upper : int
            An int describing the position of the (left/lower/right/upper) border of the
    crop's bounding box in the original image.
        """
        if left is None and right is None and upper is None and lower is None:
            self.cropped_image = image
            self.left = 0
            self.lower = 0
            self.right = image.shape[2]  # width
            self.upper = image.shape[1]  # height
        else:
            self.cropped_image = image[:, lower:upper, left:right]
            self.left = left
            self.upper = upper
            self.right = right
            self.lower = lower

        self.width = self.cropped_image.shape[2]
        self.height = self.cropped_image.shape[1]
```

```
57
58          self.horizontal_center = (self.left + self.right) / 2
59          self.vertical_center = (self.lower + self.upper) / 2
60
61      def find(self, object_name: str) -> List[ImagePatch]:
62          """Returns a list of ImagePatch objects matching object_name contained in the crop
        if any are found.
63          Otherwise, returns an empty list.
64          Parameters
65          ----------
66          object_name : str
67              the name of the object to be found
68
69          Returns
70          -------
71          List[ImagePatch]
72              a list of ImagePatch objects matching object_name contained in the crop
73
74          Examples
75          --------
76          >>> # return the foo
77          >>> def execute_command(image) -> List[ImagePatch]:
78          >>>     image_patch = ImagePatch(image)
79          >>>     foo_patches = image_patch.find("foo")
80          >>>     return foo_patches
81          """
82          return find_in_image(self.cropped_image, object_name)
83
84      def exists(self, object_name: str) -> bool:
85          """Returns True if the object specified by object_name is found in the image, and
        False otherwise.
86          Parameters
87          -------
88          object_name : str
89              A string describing the name of the object to be found in the image.
90
91          Examples
92          -------
93          >>> # Are there both foos and garply bars in the photo?
94          >>> def execute_command(image)->str:
95          >>>     image_patch = ImagePatch(image)
96          >>>     is_foo = image_patch.exists("foo")
97          >>>     is_garply_bar = image_patch.exists("garply bar")
98          >>>     return bool_to_yesno(is_foo and is_garply_bar)
99          """
100         return len(self.find(object_name)) > 0
101
102     def verify_property(self, object_name: str, visual_property: str) -> bool:
103         """Returns True if the object possesses the visual property, and False otherwise.
104         Differs from 'exists' in that it presupposes the existence of the object specified
        by object_name, instead checking whether the object possesses the property.
105         Parameters
106         -------
107         object_name : str
108             A string describing the name of the object to be found in the image.
109         visual_property : str
110             A string describing the simple visual property (e.g., color, shape, material) to
         be checked.
111
112         Examples
113         -------
114         >>> # Do the letters have blue color?
115         >>> def execute_command(image) -> str:
116         >>>     image_patch = ImagePatch(image)
117         >>>     letters_patches = image_patch.find("letters")
118         >>>     # Question assumes only one letter patch
119         >>>     return bool_to_yesno(letters_patches[0].verify_property("letters", "blue"))
120         """
```

```
121         return verify_property(self.cropped_image, object_name, property)
122
123     def best_text_match(self, option_list: List[str]) -> str:
124         """Returns the string that best matches the image.
125         Parameters
126         -------
127         option_list : str
128             A list with the names of the different options
129         prefix : str
130             A string with the prefixes to append to the options
131
132         Examples
133         -------
134         >>> # Is the foo gold or white?
135         >>> def execute_command(image)->str:
136         >>>     image_patch = ImagePatch(image)
137         >>>     foo_patches = image_patch.find("foo")
138         >>>     # Question assumes one foo patch
139         >>>     return foo_patches[0].best_text_match(["gold", "white"])
140         """
141         return best_text_match(self.cropped_image, option_list)
142
143     def simple_query(self, question: str = None) -> str:
144         """Returns the answer to a basic question asked about the image. If no question is
    provided, returns the answer
145         to "What is this?". The questions are about basic perception, and are not meant to
    be used for complex reasoning
146         or external knowledge.
147         Parameters
148         -------
149         question : str
150             A string describing the question to be asked.
151
152         Examples
153         -------
154
155         >>> # Which kind of baz is not fredding?
156         >>> def execute_command(image) -> str:
157         >>>     image_patch = ImagePatch(image)
158         >>>     baz_patches = image_patch.find("baz")
159         >>>     for baz_patch in baz_patches:
160         >>>         if not baz_patch.verify_property("baz", "fredding"):
161         >>>             return baz_patch.simple_query("What is this baz?")
162
163         >>> # What color is the foo?
164         >>> def execute_command(image) -> str:
165         >>>     image_patch = ImagePatch(image)
166         >>>     foo_patches = image_patch.find("foo")
167         >>>     foo_patch = foo_patches[0]
168         >>>     return foo_patch.simple_query("What is the color?")
169
170         >>> # Is the second bar from the left quuxy?
171         >>> def execute_command(image) -> str:
172         >>>     image_patch = ImagePatch(image)
173         >>>     bar_patches = image_patch.find("bar")
174         >>>     bar_patches.sort(key=lambda x: x.horizontal_center)
175         >>>     bar_patch = bar_patches[1]
176         >>>     return bar_patch.simple_query("Is the bar quuxy?")
177         """
178         return simple_query(self.cropped_image, question)
179
180     def compute_depth(self):
181         """Returns the median depth of the image crop
182         Parameters
183         ----------
184         Returns
185         -------
186         float
```

```
187            the median depth of the image crop
188
189        Examples
190        --------
191        >>> # the bar furthest away
192        >>> def execute_command(image)->ImagePatch:
193        >>>     image_patch = ImagePatch(image)
194        >>>     bar_patches = image_patch.find("bar")
195        >>>     bar_patches.sort(key=lambda bar: bar.compute_depth())
196        >>>     return bar_patches[-1]
197        """
198        depth_map = compute_depth(self.cropped_image)
199        return depth_map.median()
200
201    def crop(self, left: int, lower: int, right: int, upper: int) -> ImagePatch:
202        """Returns a new ImagePatch cropped from the current ImagePatch.
203        Parameters
204        -------
205        left, lower, right, upper : int
206            The (left/lower/right/upper)most pixel of the cropped image.
207        -------
208        """
209        return ImagePatch(self.cropped_image, left, lower, right, upper)
210
211    def overlaps_with(self, left, lower, right, upper):
212        """Returns True if a crop with the given coordinates overlaps with this one,
213        else False.
214        Parameters
215        ----------
216        left, lower, right, upper : int
217            the (left/lower/right/upper) border of the crop to be checked
218
219        Returns
220        -------
221        bool
222            True if a crop with the given coordinates overlaps with this one, else False
223
224        Examples
225        --------
226        >>> # black foo on top of the qux
227        >>> def execute_command(image) -> ImagePatch:
228        >>>     image_patch = ImagePatch(image)
229        >>>     qux_patches = image_patch.find("qux")
230        >>>     qux_patch = qux_patches[0]
231        >>>     foo_patches = image_patch.find("black foo")
232        >>>     for foo in foo_patches:
233        >>>         if foo.vertical_center > qux_patch.vertical_center
234        >>>             return foo
235        """
236        return self.left <= right and self.right >= left and self.lower <= upper and self.
    upper >= lower
237
238
239 class VideoSegment:
240    """A Python class containing a set of frames represented as ImagePatch objects, as well
       as relevant information.
241    Attributes
242    ----------
243    video : torch.Tensor
244    A tensor of the original video.
245    start : int
246    An int describing the starting frame in this video segment with respect to the original
       video.
247    end : int
248    An int describing the ending frame in this video segment with respect to the original
       video.
249    num_frames->int
250    An int containing the number of frames in the video segment.
```

```
251
252      Methods
253      -------
254      frame_iterator ->Iterator[ImagePatch]
255      trim(start, end)->VideoSegment
256      Returns a new VideoSegment containing a trimmed version of the original video at the [
         start, end] segment.
257      select_answer(info, question, options)->str
258      Returns the answer to the question given the options and additional information.
259      """
260
261      def __init__(self, video: torch.Tensor, start: int = None, end: int = None, parent_start
         =0, queues=None):
262          """Initializes a VideoSegment object by trimming the video at the given [start, end]
              times and stores the
263          start and end times as attributes. If no times are provided, the video is left
             unmodified, and the times are
264          set to the beginning and end of the video.
265
266          Parameters
267          -------
268          video : torch.Tensor
269          A tensor of the original video.
270          start : int
271          An int describing the starting frame in this video segment with respect to the
             original video.
272          end : int
273          An int describing the ending frame in this video segment with respect to the
             original video.
274          """
275
276          if start is None and end is None:
277              self.trimmed_video = video
278              self.start = 0
279              self.end = video.shape[0] # duration
280          else:
281              self.trimmed_video = video[start:end]
282          if start is None:
283              start = 0
284          if end is None:
285              end = video.shape[0]
286          self.start = start + parent_start
287          self.end = end + parent_start
288
289          self.num_frames = self.trimmed_video.shape[0]
290
291      def frame_iterator(self) -> Iterator[ImagePatch]:
292          """Returns an iterator over the frames in the video segment."""
293          for i in range(self.num_frames):
294              yield ImagePatch(self.trimmed_video[i], self.start + i)
295
296      def trim(self, start: Union[int, None] = None, end: Union[int, None] = None) ->
         VideoSegment:
297          """Returns a new VideoSegment containing a trimmed version of the original video at
             the [start, end]
298          segment.
299
300          Parameters
301          ----------
302          start : Union[int, None]
303          An int describing the starting frame in this video segment with respect to the
             original video.
304          end : Union[int, None]
305          An int describing the ending frame in this video segment with respect to the
             original video.
306
307          Examples
308          --------
```

```
309          >>> # Return the second half of the video
310          >>> def execute_command(video):
311          >>> video_segment = VideoSegment(video)
312          >>> video_second_half = video_segment.trim(video_segment.num_frames // 2,
     video_segment.num_frames)
313          >>> return video_second_half
314          """
315          if start is not None:
316              start = max(start, 0)
317          if end is not None:
318              end = min(end, self.num_frames)
319
320          return VideoSegment(self.trimmed_video, start, end, self.start)
321
322      def select_answer(self, info: dict, question: str, options: List[str]) -> str:
323          return select_answer(self.trimmed_video, info, question, options)
324
325      def __repr__(self):
326          return "VideoSegment({}, {})".format(self.start, self.end)
327
328      def simple_query(self, question) -> str:
329          """Ask a simple question about the video.
330
331          Examples
332          --------
333          # why does X happen?
334          # possible_answers: ['answer1', 'answer2', 'answer3', 'answer4', 'answer5']
335          def execute_command(video, question, possible_answers)->[str, dict]:
336              # Create a video segment object
337              video_segment = VideoSegment(video)
338              # The question is simple, so just ask
339              info = video_segment.simple_query("why does X happen?")
340              # Choose the answer among given possible answers
341              answer = select_answer(info, question, possible_answers)
342              return answer
343          """
344          answer = simple_query(question)
345          return answer
346
347
348  def best_image_match(list_patches: List[ImagePatch], content: List[str], return_index=False)
         -> Union[ImagePatch, int]:
349      """Returns the patch most likely to contain the content.
350      Parameters
351      ----------
352      list_patches : List[ImagePatch]
353      content : List[str]
354          the object of interest
355      return_index : bool
356          if True, returns the index of the patch most likely to contain the object
357
358      Returns
359      -------
360      int
361          Patch most likely to contain the object
362      """
363      return best_image_match(list_patches, content, return_index)
364
365
366  def distance(patch_a: ImagePatch, patch_b: ImagePatch) -> float:
367      """
368      Returns the distance between the edges of two ImagePatches. If the patches overlap, it
     returns a negative distance
369      corresponding to the negative intersection over union.
370
371      Parameters
372      ----------
373      patch_a : ImagePatch
```

```
374        patch_b : ImagePatch
375
376        Examples
377        --------
378        # Return the qux that is closest to the foo
379        >>> def execute_command(image):
380        >>>     image_patch = ImagePatch(image)
381        >>>     qux_patches = image_patch.find('qux')
382        >>>     foo_patches = image_patch.find('foo')
383        >>>     foo_patch = foo_patches[0]
384        >>>     qux_patches.sort(key=lambda x: distance(x, foo_patch))
385        >>>     return qux_patches[0]
386        """
387        return distance(patch_a, patch_b)
388
389
390 def bool_to_yesno(bool_answer: bool) -> str:
391     return "yes" if bool_answer else "no"
392
393
394 def select_answer(info: str, question: question, possible_answers: str) -> str:
395     """Given an info, question and possible answers, select the correct answer.
396
397     Examples
398     --------
399     # what does man do at the end of the video
400     # possible_answers: ['answer1', 'answer2', 'answer3', 'answer4', 'answer5']
401     def execute_command(video, question, possible_answers)->[str, dict]:
402         # Create a video segment object
403         video_segment = VideoSegment(video)
404         # Caption last frame of the video (end of video)
405         last_frame = ImagePatch(video_segment, -1)
406         last_caption = last_frame.simple_query("What is this?")
407         men = last_frame.find("man")
408         if len(men) == 0:
409             men = [last_frame]
410         man = men[0]
411         man_action = man.simple_query("What is the man doing?")
412         # Answer the question. Remember to create the info dictionary
413         info = {
414             "Caption of last frame": last_caption,
415             "Man looks like he is doing": man_action
416         }
417         answer = video_segment.select_answer(info, question, possible_answers)
418         return answer, info
419     """
420
421
422 INSERT_IN_CONTEXT_EXAMPLES_HERE
423
424
425 Write a function using Python and the VideoSegment class (above) that could be executed to
         provide an answer to the query.
426
427 Consider the following guidelines:
428 - Use base Python (comparison, sorting) for basic logical operations, left/right/up/down,
         math, etc.
429
430 INSERT_PREVIOUS_CODE_AND_ERROR_HERE
431
432 # INSERT_QUERY_HERE
```

### A.4.4   NExT-QA - Abstract API

```python
import math

class VideoSegment:
    """A Python class containing a video, as well as relevant information.
    Attributes
    ----------
    video : np.ndarray
    A tensor of the original video.
    start : int
    An int describing the starting frame in this video segment with respect to the original
    video.
    end : int
    An int describing the ending frame in this video segment with respect to the original
    video.
    num_frames->int
    An int containing the number of frames in the video segment.

    Methods
    -------
    trim(start, end)->VideoSegment
    Returns a new VideoSegment containing a trimmed version of the original video at the [
    start, end] segment.
    select_answer(info, question, possible_answers)->str
    Returns the answer to the question given the possible answers and additional information
    .
    """

    def __init__(self, video: np.ndarray, start: int = None, end: int = None, parent_start
    =0, queues=None):
        """Initializes a VideoSegment object by trimming the video at the given [start, end]
         times and stores the
        start and end times as attributes. If no times are provided, the video is left
    unmodified, and the times are
        set to the beginning and end of the video.

        Parameters
        -------
        video : np.ndarray
        A tensor of the original video.
        start : int
        An int describing the starting frame in this video segment with respect to the
    original video.
        end : int
        An int describing the ending frame in this video segment with respect to the
    original video.
        """

        if start is None and end is None:
            self.trimmed_video = video
            self.start = 0
            self.end = video.shape[0] # duration
        else:
            self.trimmed_video = video[start:end]
        if start is None:
            start = 0
        if end is None:
            end = video.shape[0]
        self.start = start + parent_start
        self.end = end + parent_start

        self.num_frames = self.trimmed_video.shape[0]

    def trim(self, start: Union[int, None] = None, end: Union[int, None] = None) ->
    VideoSegment:
```

```
56          """Returns a new VideoSegment containing a trimmed version of the original video at
    the [start, end]
57          segment.
58
59          Parameters
60          ----------
61          start : Union[int, None]
62          An int describing the starting frame in this video segment with respect to the
    original video.
63          end : Union[int, None]
64          An int describing the ending frame in this video segment with respect to the
    original video.
65
66          Examples
67          --------
68          >>> # Return the second half of the video
69          >>> def execute_command(video):
70          >>>     video_segment = VideoSegment(video)
71          >>>     video_second_half = video_segment.trim(video_segment.num_frames // 2,
    video_segment.num_frames)
72          >>>       return video_second_half
73          """
74          if start is not None:
75              start = max(start, 0)
76          if end is not None:
77              end = min(end, self.num_frames)
78
79          return VideoSegment(self.trimmed_video, start, end, self.start)
80
81      def get_video_segment_of_event(self, event) -> VideoSegment:
82          return get_video_segment_of_event(event)
83
84      def get_video_segment_before_event(self, event) -> VideoSegment:
85          return get_video_segment_before_event(event)
86
87      def get_video_segment_after_event(self, event) -> VideoSegment:
88          return get_video_segment_after_event(event)
89
90      def caption_video(self, question) -> str:
91          return caption_video(question)
92
93      def simple_query(self, question) -> str:
94          """Ask a simple question about the video.
95
96          Examples
97          --------
98          # why does X happen?
99          # possible_answers: ['answer1', 'answer2', 'answer3', 'answer4', 'answer5']
100         def execute_command(video, question, possible_answers)->[str, dict]:
101             # Create a video segment object
102             video_segment = VideoSegment(video)
103             # The question is simple, so just ask
104             info = video_segment.simple_query("why does X happen?")
105             # Choose the answer among given possible answers
106             answer = select_answer(info, question, possible_answers)
107             return answer
108         """
109         answer = simple_query(question)
110         return answer
111
112
113 def select_answer(info: str, question: question, possible_answers: str) -> str:
114     """Given an info, question and possible answers, select the correct answer.
115
116     Examples
117     --------
118     # what does person A do after event X?
119     # possible_answers: ['answer1', 'answer2', 'answer3', 'answer4', 'answer5']
```

```
120    def execute_command(video, question, possible_answers)->[str, dict]:
121        # Create a video segment object
122        video_segment = VideoSegment(video)
123        # Get video segment after event X
124        video_segment_after = video_segment.get_video_segment_after_event("event X")
125        # Ask what the person A is doing
126        info = video_segment_after.caption_video("What is person A doing?")
127        # Choose the answer among given possible answers
128        answer = select_answer(info, question, possible_answers)
129        return answer
130    """

132
133  INSERT_IN_CONTEXT_EXAMPLES_HERE

135  Write a function using Python and the VideoSegment class (above) that could be executed to
           provide an answer to the query.

137  Consider the following guidelines:
138  - Use base Python (comparison, sorting) for basic logical operations, left/right/up/down,
           math, etc.
139  - The input to your program is a video, question and possible answers.
140  - Always start your function by creating a 'video_segment = VideoSegment(video)' object.

142  INSERT_PREVIOUS_CODE_AND_ERROR_HERE

144  # INSERT_QUERY_HERE
```

