# OpenReview forum: "Towards Truly Zero-shot Compositional Visual Reasoning with LLMs as Programmers"
_TMLR — Accepted by TMLR_

### Review · Reviewer_g5Cs · 2024-01-25

**Summary Of Contributions:**

This paper introduces a framework to enhance the robustness of Language Model-Led (LLM) programming for visual reasoning tasks.

- **Automatic Generation of Query-Code ICEs (ACEs)**: It removes the necessity for manually crafted in-context examples (ICEs), advancing the models towards genuine zero-shot visual reasoning capabilities. It demonstrates the zero-shot generation of ACEs using a few labeled examples, which, when used as in-context examples, significantly improve performance and eliminate the need for human-designed ICEs.

- **Introduction of "Abstract API"**: This API comprises spatially and temporally abstract routines, which ease the spatial and temporal reasoning load on the code-generating LLM, enhancing performance.

- **Self-Correction Capabilities in LLMs**: Showcases the ability of LLMs to perform "self-correction" through "self-debugging" and "self-tuning" methods without requiring ground-truth labels. (1) Self-Debugging: Generating new code from scratch yields better results than using previous query-code pairs as feedback. (2) Self-Tuning: Highlights the automatic tuning of hyperparameters in certain modules if code execution fails.

- **Performance Improvement Across Tasks**: Validates that each component of the proposed framework consistently enhances performance in various compositional question-answering and video temporal reasoning tasks.

**Audience:**

Yes

**Broader Impact Concerns:**

Accessibility and Inclusivity: It will increase the impact of this paper if the system's advanced capabilities can be accessible to a diverse user base, including those with limited resource. This work considers several private APIs or high cost models, including PaLM 2 (Anil et al., 2023), OWLv2 (Minderer et al., 2023), PaLI-3 (Chen et al., 2023d). It naturally raise the concerns how the cost is affordable for many university labs. Ensuring the technology is inclusive is an important ethical consideration.

**Claims And Evidence:**

Yes

**Requested Changes:**

**Addressing Limited Impact**

- Comparative Analysis: Conduct a comparative study of the proposed method's effectiveness and generality against more recent methods like Visual ChatGPT (Wu et al., 2023), MM-ReAct (Yang et al., 2023b), and LLaVA-Plus (Liu et al, 2023). This will provide a clearer understanding of where your method stands in the current landscape of LLM and LMM advancements.

- Method Integration: Explore the possibility of integrating or chaining your method with these recent multimodal tools. This could potentially enhance the capabilities and applicability of your framework in the broader context of LLM and LMM research.

**Updating Missing Literature**

- Literature Review Expansion: Update the literature review section to include recent developments in LMM research, particularly those outlined in "A Survey on Multimodal Large Language Models" and "Multimodal foundation models: from specialists to general-purpose assistants." This will help position your work within the current state of the field.

- Contextual Relevance: Re-evaluate and describe your work in the context of these latest developments. Highlight how your method contributes to or differs from the current trends in LMM.

**Mitigating Technical Limitations**

- Scalability Assessment: Conduct a thorough assessment of the scalability of your approach. Include tests or simulations that demonstrate how your framework performs with an increasing number of self-tuning and self-debugging calls, external tools or in more complex reasoning scenarios. Based on these assessments, provide recommendations or modifications to improve scalability.

- Technical Documentation: Offer detailed technical documentation outlining the computational requirements and practical limitations of the framework. This will help users understand the necessary resources and potential constraints when implementing your method.

These changes aim to enhance the relevance, applicability, and practical utility of your research, ensuring it aligns with the latest developments in the field and addresses key technical challenges.

**Strengths And Weaknesses:**

**Strengths of the Submission**

- *Zero-Shot Learning Enhancement*: The paper addresses an important  problem in tool use of multimodal LLM, how to reduce the human burden in creating the in-context examples (ICEs). The elimination of the need for human-engineered  ICEs and the introduction of automatic generation of query-code ICEs (ACEs) is a substantial stride towards true zero-shot learning.

- *The method is clearly described and well motivated*: The implementation of the "Abstract API" effectively reduces the complexity of spatial and temporal reasoning for LLMs, thereby improving the overall robustness of the model. The paper's exploration of self-correction through self-debugging and self-tuning is a novel concept, enhancing the model's reliability and reducing dependence on external inputs.

- *Empirical Validation*: The comprehensive testing across various tasks demonstrates the practical effectiveness of the proposed framework in general and ablates the impact of each components.


**Weaknesses That Require Attention**
-  *Limited impact*: Though the proposed method improve ViperGPT, there are several more effective methods for LLM or large multimodal model (LMM) to leverage external tools/API, for example, chaining LLM with multimodal tools such as Visual ChatGPT (Wu et al., 2023), MM-ReAct (Yang et al., 2023b), and training LMM with multimodal tools such as LLaVA-Plus (Liu et al, 2023). It would great to study the effectiveness and generality of the the proposed method with some of these more recent methods.

-  *Missing Literature*: There is an emerge of LMM research and papers in the late half of 2023. The paper's focus seems a bit outdated. Please see the some related survey papers [1,2], and describe the proposed work in the context of up-to-date LMM development.

[1] A Survey on Multimodal Large Language Models

[2] Multimodal foundation models: from specialists to general-purpose assistants

-  *Technical limitations*: (1) Resource Intensity. The framework's complexity, particularly in terms of self-tuning and self-debugging processes, might entail significant computational resources, which could be a barrier in resource-constrained environments. (2) Scalability Issues. The scalability of the proposed approach, especially when dealing with a large number of tools or more complex reasoning tasks, is not clearly addressed, which could be a concern for practical applications.

---

> ### Author Response · Authors · 2024-03-16
>
> Dear Reviewer,
>
>
> Thank you for your review. We greatly appreciate your thoughtful and detailed review and the time you've taken to evaluate our work. Your feedback helps us refine and improve our work.
>
>
> We thank you for noting that our paper addresses an “important problem” and that we make “substantial strides towards true zero-shot learning”, that our method is “clearly described and well motivated”, effective, that “self-debugging and self-tuning is novel concept” and that our evaluation is comprehensive.
>
>
> Below are our responses to the points you raised, questions and requested changes:
>
>
> **Limited impact:**
> > *proposed method improves ViperGPT, but there are several more effective methods (ChatGPT, MM-ReAct, LLaVa-Plus).. study the effectiveness and generality of the proposed method with more recent methods *
>
>
> We thank the reviewer for pointing these works out. We added all of them to the related work section, as well as many other recent papers from the surveys that were pointed out to us. Unfortunately, VisualChatGPT and MM-React do not perform any quantitative analysis in their reports, do not propose systematic evaluation scenarios for their methods, and therefore seem to be documentation of demos, rather than scientific papers. Moreover, we could not find any quantitative results in the papers citing Visual ChatGPT and MM-ReAct, so it is unclear what evidence the reviewer bases their claim of these methods being “more effective” on. LLaVA-Plus, on the other hand, learns to use tools by end-to-end training, which is different to the ViperGPT setup that does not need any gradient updates, so a quantitative comparison would not be meaningful.
>
>
> **Missing Literature:**
>
>
> We thank the reviewer for pointing out these surveys. We added missing literature to the related work section in the revised version of our manuscript. We also added references to the surveys that the reviewer suggested. Please let us know in case there are any particular papers that you think we missed and we would be happy to update our paper.
>
>
> **Technical limitations:**
> > *Resource Intensity and Scalability*
>
>
> Although we agree that approaches based on “tool use” tend to be computationally demanding, we note that this is the case for all such approaches, and not specific to our framework. In particular for our approach of “self-debugging” only the code-generating LLM will be executed a few times, all other models only once (when the code compiles). Therefore, the computational overhead is limited.
> Similarly, ACEs are only generated *once* per dataset, so there is no computational overhead during inference time.
>
>
> -------------------------------------------------------
> ## Requested Changes
>
>
> **Comparative Analysis with recent methods**
>
>
> Please see our response to the same comment in the “weaknesses” above.
>
>
> **Method Integration and Contextual Relevance**
> > *possibility of integrating or chaining your method with the recent methods*
>
>
> We believe integration of our framework with these new multimodal tools would be interesting, however we leave this for future work as it would require a disproportionate amount of work.
>
>
> **Literature Review Expansion:**
>
>
> As noted above, we updated our literature in line with your proposal.
>
>
> **Scalability Assessment:**
> > *assess the scalability of the approach.. provide recommendations to improve scalability.*
>
>
> First of all, we would like to note that our framework is *model agnostic*. Therefore, the runtimes will heavily depend on the choice of LLM we use for code generation and on the hardware we use for their deployment.
> In self-debugging, only the code-generating LLM is called multiple times, whereas vision modules are called only once (when the code executes correctly). Therefore, this approach scales linearly with the LLM execution time. In self-tuning, only the vision modules are called multiple times, whereas the code-generating LLM is executed only once. Therefore, this approach scales linearly with the execution time of the vision modules (this time is sample/question-dependent). Lastly, with certain optimizations, such as batching of the calls to the vision/language modules (that can be done on the server side), we can get significant performance gains.
>
>
> **Technical Documentation:**
> > *computational requirements and practical limitations of the framework*
>
>
> Please note that our framework requires only negligible computation compared to the LLMs and pretrained vision models that it uses. Therefore, the computational requirements are heavily tied to the choice of these models and orthogonal to our proposed improvements.

---

### Review · Reviewer_5AHJ · 2024-01-29

**Summary Of Contributions:**

This work introduces a framework enhancing visual reasoning in large language models (LLMs) as controllers. Traditional end-to-end neural networks usually falter in compositional reasoning, generalization, and fine-grained spatial-temporal reasoning. Their approach addresses these challenges by decomposing tasks into subtasks with a set of visual tools for solutions. They propose an "Abstract API" for spatial and temporal abstraction, reducing reliance on the LLM's intrinsic reasoning capabilities. Additionally, Their framework automatically generates in-context examples using a small set of labeled instances, eliminating the need for labor-intensive, human-crafted examples. Addtionally, it enables models to perform self-correction through self-debugging and self-tuning mechanisms. In some common visual reasoning tasks, including visual grounding, compositional image question answering, and video temporal reasoning, the experimental results show the strength of the proposed method.

**Audience:**

Yes

**Broader Impact Concerns:**

The proposed technicals in this paper have no significant concerns on the ethical aspects.

**Claims And Evidence:**

Yes

**Requested Changes:**

- For Automatic generation of in-context examples, the current way starts from a small set of labeled examples as shown in Section 2.3. Could the requirement for a small set of labeled examples be eliminated?
- Can the author try GPT-4 to see if the proposed abstract API can still bring improvements?
- Tuning hyper-parameter is not exactly self-tuning to me. Are there further powerful or fundamental way here for self-tuning?
- For verifying the effectiveness of the proposed module, can the author provide some qualitative comparisons?

**Strengths And Weaknesses:**

The studied direction, visual reasoning, is important, and the authors did several good attempts. Overall the proposed components are reasonable and are composed in a sound way, including abstract API, ACE, and self-correction mechanism. The paper is easy to follow.

The author conducted extensive experiments in standard datasets including, RefCOCO, RefCOCO+, GQA, and NExT-QA covering three major visual reasoning tasks (visual grounding, compositional image question answering, and video temporal reasoning). Also, extensive quantitatively ablation studies are conducted to verify the effectiveness of the proposed modules..

I have several major concerns the below:
- The current design of Abstract API including get_patch_left_of, get_patch_right_of, get_patch_above_of, get_patch_below_of, and more, seems pretty ad-hoc. I expect advanced LLM and VLM and learn to propose them naturally or by asking them firstly to propose those high-level spatial-temporal operators.
- I am unsure the significance of proposing Automatic generation of in-context examples (Section 2.3). This seems like a trivial solution to me.
- How does the self-correction (Section 2.4) different to previous literatures such as Reflexion and self-improve llm papers? Only can tune hyperparameters seems not a substantial difference.

---

> ### Author Response · Authors · 2024-03-16
>
> Dear Reviewer,
>
>
> Thank you for your review. We appreciate your insights and the time you've taken to evaluate our work.
>
>
> Thank you also for noting that the “studied direction is important”, that the “components of our approach are reasonable and composed in a sound way”, that the paper is “easy to follow” and that our “experiments are extensive”.
>
>
> Below, we address the points you raised.
>
>
> **Abstract API Design**
> > *current design of Abstract API … seems pretty ad-hoc*
>
> Thank you for your comment, however we would like to respectfully disagree. The design of Abstract API is not ad-hoc at all. It is a priori unclear whether an array listing objects in a certain order e.g. in the x-direction, is ordered from left to right or the opposite way. This is a clear failure mode we observed in ViperGPT, which motivated the introduction of the Abstract API. This can be considered as closing gaps in visual grounding with structured prompting via API rather than ad-hoc in natural language. Note also that in ViperGPT the authors provide a number of code examples that instruct the model how to handle failure modes (e.g. “chair at the front”, “car just under stop sign”, “middle kid”, “balloon on the right and second from the bottom” and many other human-engineered ICEs: https://github.com/cvlab-columbia/viper/blob/main/prompts/benchmarks/refcoco.prompt). These examples are dataset-specific and thus far less general than the Abstract API we propose.
>
>
> **ACEs significance:**
> > *..unsure about the significance of proposing Automatic generation of ICEs.. seems like a trivial solution to me..*
>
> From the LLM literature, we know that in-context examples (ICEs) bring large gains. However, creating these requires significant human expertise. Our proposed automatic generation of ICEs (ACEs) generates these examples automatically. Its significance is therefore in automatizing large parts of high-skill human labor (i.e. programming). While conceptually simple, this has not been proposed before, and we show that it is quite effective in our extensive experiments. We hope it finds wide applicability thanks to its simplicity.
>
>
> **self-correction relation to previous literatures such as Reflexion**
>
>
> Thank you, we actually already cited Reflexion in our related work section. Our method differs from Reflexion in several ways. First of all, we perform self-correction without *any* ground-truth data, unlike Reflexion which uses ground-truth data. We simply check if the generated code compiled and executed. Therefore we perform an “evaluation” step without any labels, which cannot be done in Reflexion and similar approaches. Reflexion further employs CoT to generate unit tests that are specific for programming tasks. Our approach is general and not specific to any task. Finally, Reflexion considers tasks such as natural language question-answering, programming and decision making, whereas we present self-correction for visual QA.
>
>
>
>
> ----
> ### Requested changes
>
>
> **Eliminate requirement for a small set of labeled examples:**
> > *Could the requirement for a small set of labeled examples (in ACEs) be eliminated?*
>
>
> Our current setup requires a small set of labeled examples. Note that this requires significantly less effort compared to prior work where humans write ICEs (specifically programs) manually. This is also a standard requirement in few-shot learning literature. The only obvious alternative is prompt engineering, which in this case seems much more expensive than collecting a few labeled examples.
>
>
> **Performance with GPT-4**
>
>
> Even though we agree with the reviewer that this would be an interesting experiment, we would note that the choice of LLM is orthogonal to our contributions.  Moreover, evaluating GPT-4 would incur an additional cost and we already showed gains on two different LLMs (one from the OpenAI’s models family).
>
>
> **Other ways for self-tuning:**
>
>
> Thank you for your question. We would first like to note that tuning hyper-parameters is particularly well suited when working with visual tools. Also, this approach might be somewhat less brittle than repeated LLM calls, e.g. when feeding the LLM outputs as inputs, which has a risk of accumulating hallucinations.
> Apart from tuning hyper-parameters, there are many other ways how LLMs-as-controllers could self-tune. For example, they could learn how to change the API in the prompt, for example by adding more complex (abstract) routines. Also, they could change the task instruction in the prompt, or add more ICEs. We leave these directions for future work.
>
>
> **Qualitative comparison:**
> > *For verifying the effectiveness of the proposed module, can the author provide some qualitative comparisons?*
>
>
> The LLM and the pretrained vision modules used in our framework and the baselines are the same (our contributions are orthogonal to those). Therefore, inspecting individual samples does not provide additional insights compared to computing aggregate statistics.

---

### Review · Reviewer_dZpC · 2024-03-11

**Summary Of Contributions:**

This work builds upon the recent success of using LLMs for writing programs to solving visual reasoning tasks. These synthesized programs leverage pre-written functions that allow calling individual pre-trained computer vision models (like object detection, depth estimation, etc). These existing methods query LLMs using expert written programs for solving a few example visual reasoning questions.

This work addresses the following shortcomings:
1. The predefined functions in the existing methods focus on higher level semantic tasks like object localization, attribute classification, etc. This work introduces additional "primitive" functions. (referred to as Abstract API)
2. Existing methods include expert written example programs for the LLMs query (for in-context learning). This work proposes a solution to avoid this requirement by automatically synthesizing a few programs using ground truth (image, question, answer) triplets. (referred to as ACE)
3. Programs generated by existing methods sometimes do not compile/run due to errors. This work proposes to use the error as feedback to re-synthesize a better program. (referred to as self-correction)

**Audience:**

Yes

**Claims And Evidence:**

No

**Requested Changes:**

Overall the ideas presented are sound improvements over the implementation of the original ViperGPT. However, I think the paper claims a very broad significance of the ideas which is not necessarily backed by evidence / discussion.

I believe that since the overall ideas are effective, modifying/rephrasing the claims in the text might be sufficient. Alternatively, if I have misunderstood the significance of these ideas, please provide clarifications.

**Strengths And Weaknesses:**

# Strengths

- The paper is clearly written and is easy to follow. The introduction thoroughly motivates the problem by identifying shortcomings for existing works like ViperGPT. The method is concisely presented with examples and intuitions.
- The paper proposes three useful ideas (mentioned above). Each of these ideas will be potentially useful for researchers working on visual reasoning problems using LLMs.
- The experimental section has been well thought out. The paper evaluates the overall system using the three proposed ideas on several visual reasoning benchmarks (RefCOCO, RefCOCO+, GQA, Next-QA). The experiments also include evaluation of each individual proposal separately to demonstrate its efficacy. These experiments provide interesting insights that might be useful for researchers.

# Weaknesses:

Overall, I think the paper overstates several contributions/claims.

### Abstract API - how is it creating abstractions?

The intuition, motivation and justification for introducing an "Abstract API" doesn't seem to align with the actual implementation. More concretely, here is the justification for the Abstract API:
```
When programming, we continuously build new layers of abstraction. We start from a set of primitive
functions and then abstract them away by adding new functions. These are then grouped into libraries, and
additional layers of abstraction are built on top. By abstracting away the implementation details, we are able
to build systems with ever-increasing capabilities and complexity.

Motivated by this, we introduce a set of
spatially and temporally abstract functions hat abstract away a number of lines of code for the
same functionality and together make the Abstract API.
```

The actual implementation of the Abstract API involves simply defining additional functions: `get_patch_left_of, get_patch_right_of, get_patch_above_of, get_patch_below_of, etc`. Its unclear to me how this is creating abstractions in the API.

These functions simply reduce the burden of the LLM. For example, the original ViperGPT would have to synthesize a couple of lines of code to find the "patch left of" but now the LLM can simply call this function. This is also clear from the following motivation presented:
```
For example, for the query “return the second person from the right”, the program generated by the LLM correctly sorts the persons along the horizontal axis but then wrongly takes the second index in the array (instead of the second last).
```

The additional functions seem to be merely identifying logical sequences that are difficult for the LLM and replacing them with hand-crafted functions in the API.


### Issues with automatic generation of in-context examples

I think the idea of ACE is generally useful and seems to work well. However, I think it requires further investigation for reliability.
The in-context samples come from LLMs trying to synthesize programs for an (image, question) pair without any context. The only quality control / filtering for these generations is the evaluation using the annotated answers. However, I think several faulty programs could lead to the same answer.

For example, an image where there are two black dogs and the question is "what is the color of the second dog from the left?". Several incorrect programs could lead to the answer "black".
The paper claims that the overall pipeline is more robust than using human crafted context. I'm not sure if this is true given the unreliability of the in-context samples.

This could be solved by performing additional analysis on the synthesized programs across several samples.


### Applicability of Self-tuning
```
In some cases, we know that code execution failed due to some components of a particular module. For example, the open vocabulary detector fails due to a too high threshold hyperparameter. When the threshold is high, we have a higher number of false negatives. For such cases, we propose to automatically change the hyperparameter of the module (e.g. reduce the threshold) and execute code again.
```
The applicability of this seems to be overstated. The text of the paper only suggests and evaluates one very specific version of this self-tuning idea - **lowering the threshold of object detectors**. There is no discussion on other avenues where this can be used and how it would be applied.

Here is another example where the paper seems to overstate the applicability:
```
In self-tuning, we show that hyper-parameters of certain modules can be tuned automatically if code execution fails due to these modules.
```
Based on the paper, there is evidence for only one module that can be tuned. I'm not sure if the results generalize to other possible applications of this idea. At the very least, there needs to be a discussion on the intuition behind why this idea might be more generally applicable.

---

> ### Author Response · Authors · 2024-03-16
>
> Dear Reviewer,
>
> Thank you for your review. We greatly appreciate your thoughtful and detailed review and the effort you've taken to evaluate our work. Your feedback helps us to refine and improve our work.
>
> We thank you for noting that the ideas presented are “sound improvements”, effective and can be useful for researchers working on visual reasoning with LLMs, that the paper is clearly written, that the method and introduced ideas are motivated well and are concisely presented, and that the experiments are “well thought out” and provide interesting insights.
>
> Below are our responses to the points you raised:
>
> **Overall**
> > *..overall ideas are effective, modifying/rephrasing the claims in the text might be sufficient..*
>
> We thank you for your feedback and we agree to make the requested changes as suggested.
>
> **Abstract API**
> > *The intuition, motivation and justification for introducing an "Abstract API" doesn't seem to align with the actual implementation...*
>
> As noted in the text you quote, we denote these functions “abstract” inspired by the fact that they “abstract away” concepts which cannot be learned from text alone and require significant overhead to learn from ACEs. Therefore, by this we primarily mean conceptually abstract functions. We also believe that such a presentation of these ideas could additionally clarify the motivation and inspire future work in this direction. We do, however, agree with the reviewer to adjust the text to better reflect the implementation. In the revised manuscript (Sec 2.2, Par.1), we rephrase the analogy with OOP and toned down claims, as the reviewer suggested.
>
> **Automatic generation of ICEs**
>
> > *idea of ACE is generally useful and seems to work well... requires further investigation for reliability... paper claims that the overall pipeline is more robust than using human crafted context. I'm not sure if this is true given the unreliability of ICEs.*
>
> We acknowledge the potential drawback of ACEs that could have incorrect programs that lead to correct solutions. However, in the paper we do not claim that ACEs make the setup *more robust* than when using human crafted ICEs (considering one specific, fixed set of LLM, API, vision tools, etc). Of course, if given the freedom to hand-write the ICEs, we not only have more guarantees that the code is correct, but we can also distill much more supervision about the dataset-specific “tricks” into the ICEs, such as “Clothing always requires returning a person” instruction in ViperGPT’s ICEs: https://github.com/cvlab-columbia/viper/blob/main/prompts/benchmarks/refcoco.prompt#L447
>
> In our paper, “robust” refers to the framework as a whole. Robustness is obtained when using ACEs compared to the case when *not* using any ICEs (assuming that we do not want to write them by hand), and when requiring the setup to be agnostic to the choice of LLM, API, vision tools etc. Even if we decide to write the examples by hand, if the LLM changes, it is highly likely we should rewrite these examples from scratch. To support this argument, note the difference in the scores we observed when running the original ViperGPT code from their GitHub repository with their API and their hand-written ICEs with GPT-3.5 instead of the Codex model that was used for the experiments in the ViperGPT paper (this difference is also noted by the authors on multiple GitHub “issues”, e.g. https://github.com/cvlab-columbia/viper/issues/44). On the other hand, ACEs can be freely generated, therefore resulting in a more “generalizable” setup assuming we do not want to rewrite ICEs from scratch every time the LLM or the API is changed.
> Additionally, we can see from the Error analysis in Figure 5 that all types of errors reduce when using ACEs, especially when combined with self-tuning and Abstract API.
>
> **Self-tuning**
>
> > *The applicability of this (self-tuning) seems to be overstated. The text of the paper only suggests and evaluates one very specific version of this self-tuning idea.. no discussion on other avenues where this can be used and how it would be applied...*
>
> Thank you for pointing this out, we added a discussion to clarify the intuition behind this idea. Indeed, although in our paper we experiment with only a single such option (object detector), the idea is applicable to any model that has “sensitivity” HPs. For example, another option could be the threshold for the “similarity” of the CLIP-style image-text model. This model filters out a particular instance (e.g. “blue ball”) from a set of instances (“balls”), e.g. based on a noun phrase (“blue ball”). It compares the text embedding of “blue ball” with image embeddings of all “ball” patches. It then returns all patches whose embeddings have similarity higher than the threshold with the text embedding. Here we would similarly tune the threshold in case this module failed. We added a discussion on this to the revised manuscript (Sec.2.4,Par.3 and Conclusion) as suggested by the reviewer.

---

### Author Response · Authors · 2024-03-16
**General Reviewer Response**

Dear Reviewers and AE,

Thank you for your reviews and all your constructive feedback.

We replied to each review individually, and uploaded a revised version of our manuscript, with the changes you suggested. We left the previous text crossed-out in red color, whereas the new text we added is in blue ink.

Thank you again and we are happy to answer any further questions you may have.

---

### Decision · Action_Editor_X28V · 2024-04-24

**Recommendation:** Accept as is

**Comment:**

The submission received three expert reviews.  Post rebuttal, all reviewers agreed that the submission has met the two TMLR acceptance criteria.  The reviewers suggested that three ideas---abstract APIs, the automatic generation of ICEs (ACEs), and self-tuning---could be interesting enough to part of the TMLR audience.  Reviewers also suggested that the submission may have limited significance and impact as there have been other similar works with stronger performance, though this does not impact the fact that the submission has met the criteria to be accepted by TMLR.

**Audience:**

Yes.

**Claims And Evidence:**

Yes, the claims are supported.